# Bayesian Neural Scaling Law Extrapolation with Prior-Data Fitted Networks

Dongwoo Lee [* 1]   Dong Bok Lee [* 2]   Steven Adriaensen [3]   Juho Lee [2]   Sung Ju Hwang [2 4]   Frank Hutter [3]
Seon Joo Kim [1]   Hae Beom Lee [5]

## Abstract

Scaling has been a major driver of recent advancements in deep learning. Numerous empirical studies have found that scaling laws often follow the power-law and proposed several variants of power-law functions to predict the scaling behavior at larger scales. However, existing methods mostly rely on point estimation and do not quantify uncertainty, which is crucial for real-world applications involving decision-making problems such as determining the expected performance improvements achievable by investing additional computational resources. In this work, we explore a Bayesian framework based on Prior-data Fitted Networks (PFNs) for neural scaling law extrapolation. Specifically, we design a prior distribution that enables the sampling of infinitely many synthetic functions resembling real-world neural scaling laws, allowing our PFN to meta-learn the extrapolation. We validate the effectiveness of our approach on real-world neural scaling laws, comparing it against both the existing point estimation methods and Bayesian approaches. Our method demonstrates superior performance, particularly in data-limited scenarios such as Bayesian active learning, underscoring its potential for reliable, uncertainty-aware extrapolation in practical applications. The code and models are available at https://github.com/DongWooLee-Eli/nslpfn.

## 1. Introduction

Recent advancements in both the vision (Tan, 2019; Alexey, 2020; Kolesnikov et al., 2020; Dai et al., 2021; Zhai et al., 2022) and natural language processing (NLP) (Devlin, 2018; Brown, 2020; Achiam et al., 2023; Touvron et al., 2023a;b;

*Equal contribution [1]Yonsei University [2]KAIST [3]University of Freiburg [4]DeepAuto [5]Korea University. Correspondence to: Hae Beom Lee <haebeomlee@korea.ac.kr>.

*Proceedings of the 42nd International Conference on Machine Learning*, Vancouver, Canada. PMLR 267, 2025. Copyright 2025 by the author(s).

Dubey et al., 2024) have largely been driven by scaling. Numerous studies have rigorously explored *neural scaling laws* (Hestness et al., 2017; Johnson et al., 2018; Rosenfeld et al., 2019; Kaplan et al., 2020; Rosenfeld, 2021; Ghorbani et al., 2021; Bansal et al., 2022; Hoffmann et al., 2022; Chung et al., 2024), finding that performance metrics, such as error rate or loss, frequently follow the power law function, as key factors such as size of data and model or training duration increase. The problem is that scaling up those key factors for modern neural architectures requires high costs (*e.g.*, collecting more labeled data or consuming more training resources for larger models), and it would be beneficial to foresee the behavior of neural scaling laws from observations at small-scale, before scaling up.

Fortunately, recent studies (Hoffmann et al., 2022; Bansal et al., 2022; Alabdulmohsin et al., 2022; Caballero et al., 2022) have shown that the neural scaling laws can be predicted empirically in modern neural architectures with carefully crafted functional forms. As typically done in regression, after predefining a reasonable functional form, the parameters of the function can be estimated with maximum likelihood given the partial observations in a neural scaling law. After the estimation, extrapolation is done to predict the future behavior of each curve. Such *neural scaling law extrapolation* not only provides the understanding of modern neural architectures (Abnar et al., 2021; Bansal et al., 2022; Bahri et al., 2024), but is also applicable to a range of fields; such as determining the proper amount of samples in data-scarce domains, *e.g.*, medicine (Mukherjee et al., 2003; Figueroa et al., 2012; Beleites et al., 2013; Cho et al., 2015), enabling the early-stopping of the hyperparameter optimization (Domhan et al., 2015; Hestness et al., 2017; Johnson et al., 2018), or neural architecture search (Hestness et al., 2017; Elsken et al., 2019; Klein et al., 2022).

However, despite the recent success on the neural scaling law extrapolation (Hoffmann et al., 2022; Bansal et al., 2022; Alabdulmohsin et al., 2022; Caballero et al., 2022), the existing works are inherently limited as they do not consider estimating *uncertainty*. Considering that the true potential of neural scaling law extrapolation comes from utilizing it to make important decisions such as whether to spend computational costs further or not, naively relying on simple point estimation is too risky, as depicted in Fig. 1. Therefore, in

this paper, we stress the need for a Bayesian approach to capture uncertainty and quantify the reliability of neural scaling law extrapolation. However, typical Bayesian methods, like directly applying Markov Chain Monte Carlo (MCMC) on existing functional forms, struggle to handle chaotic behaviors—such as non-monotonicity—that are prevalent in real-world neural scaling laws (Nakkiran et al., 2021). This is because not only are these optimization landscapes non-convex or multi-modal (Caballero et al., 2022) but also setting a practical prior distribution that can cover these chaotic behaviors is challenging.

To overcome this challenge, we investigate the potentials of Prior-data Fitted Networks (PFNs; Müller et al., 2021) for neural scaling law extrapolation. PFNs are an in-context approximate Bayesian inference method, which is meta-learned with the training data sampled from the prior distribution. PFNs provide several benefits over MCMC. Specifically, it is very flexible in defining any complex prior distributions as long as we can efficiently sample from them. Further, the speed of inference is faster by multiple orders of magnitudes since PFN is an in-context inference method. We empirically validate the efficacy and efficiency of our approach, which we name as **Neural Scaling Law PFN** (**NSL-PFN**), on an extensive set of datasets by comparing it not only with the recent point estimation methods but also their MCMC variants. We also show that our NSL-PFN reliably predicts chaotic behaviors of real-world neural scaling laws even with a few observations at a small scale.

We summarize the contribution of our paper as follows:

- To our knowledge, we introduce a Bayesian method for extrapolating neural scaling laws for the first time.

- We propose a novel prior distribution for PFN which is specifically tailored to neural scaling law extrapolation.

- We empirically validate that our method provides better point estimate fits with the ability to automatically infer the best functional form and the number of breaks.

- We empirically show that our method provides better uncertainty than the non-specialized PFNs and other MCMC baselines, on many real-world neural scaling law examples as well as Bayesian active learning settings.

## 2. Background and Related Work

### 2.1. Point Estimation of Neural Scaling Laws

$\mathcal{M}_1, \mathcal{M}_2$, and $\mathcal{M}_3$. Most existing methods for estimating neural scaling laws rely on point estimation. The power law function is the most representative function used for extrapolation and is known to work well for a wide set of neural scaling laws, including many vision and natural language processing tasks (Hestness et al., 2017; Johnson et al., 2018; Rosenfeld et al., 2019; Kaplan et al., 2020; Rosenfeld,

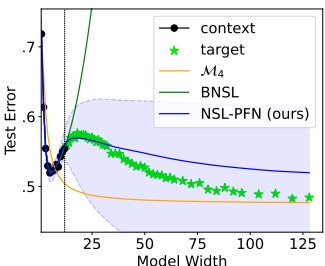

Figure 1: **Extrapolation results of each method on a neural scaling law from the double descent dataset (Nakkiran et al., 2021).** Here, the context and target denote the observations and the target points we want to correctly estimate, respectively.

2021; Ghorbani et al., 2021; Bansal et al., 2022; Hoffmann et al., 2022; Chung et al., 2024). Let $y$ denote the performance such as prediction error or cross-entropy, and $x$ the quantity that is being scaled such as the size of dataset or the number of parameters in the neural network. The power law function in its simplest form is given by:

$$\mathcal{M}_1: \quad y = ax^{-b}, \tag{1}$$

$$\mathcal{M}_2: \quad y = ax^{-b} + c, \tag{2}$$

where $a, b, c \geq 0$ are the coefficients we want to estimate. The only difference between $\mathcal{M}_1$ and $\mathcal{M}_2$ is whether there is a bias $c$ or not for capturing saturating performance (Mukherjee et al., 2003; Figueroa et al., 2012; Domhan et al., 2015; Hestness et al., 2017; Johnson et al., 2018; Rosenfeld et al., 2019; Gordon et al., 2021). For neural machine translation, Bansal et al. (2022) propose a slightly more expressive form by adding a shifting term $d \geq 0$ to $\mathcal{M}_1$:

$$\mathcal{M}_3: y = a(x^{-1} + d)^b. \tag{3}$$

Note that $\mathcal{M}_3$ reduces to $\mathcal{M}_1$ when $d = 0$.

$\mathcal{M}_4$. However, those $\mathcal{M}_1$, $\mathcal{M}_2$, and $\mathcal{M}_3$ functions do not have any inflection points, which frequently appear in many scaling law examples—initially the scaling law curve is flat, *i.e.*, close to the initial performance $y_0$, then gradually starts following a power law function. Alabdulmohsin et al. (2022) propose the following function family to express such inflection points:

$$\mathcal{M}_4: y = g^{-1}(x), \quad \text{where}$$
$$x = g(y) = \left( \frac{y - c}{a(y_0 - y)^\alpha} \right)^{-\frac{1}{b}}, \tag{4}$$

where $a, b, \alpha \geq 0$ and $y_0 \geq c \geq 0$. Similarly to $\mathcal{M}_3$, this $\mathcal{M}_4$ function reduces to $\mathcal{M}_2$ when $\alpha = 0$ (since we have $y = ax^{-b} + c$). Therefore, the role of $\alpha$ is to control the degree to which the scaling law initially sticks to $y_0$.

**Broken Neural Scaling Laws (BNSL).** While $\mathcal{M}_4$ is more flexible than $\mathcal{M}_1$, $\mathcal{M}_2$, and $\mathcal{M}_3$, it still has a limitation that the function is defined as an inverse—the function

should be bijective. However, some real-world scaling law examples exhibit more complicated behaviors such as *double descent* (Nakkiran et al., 2021)—the curve initially decreases, then suddenly increases, and then decreases again eventually following a power law function. Thus, Caballero et al. (2022) propose the following function family:

$$\textbf{BNSL: } y = c + ax^{-b} \prod_{i=1}^{n} \left( 1 + \left( \frac{x}{e_i} \right)^{1/s_i} \right)^{-\Delta b_i \cdot s_i}. \quad (5)$$

Eq. (5) is nothing but the smooth piece-wise concatenation of power law functions, where $n$ is the number of *breaks*, making the corresponding curve consists of $n + 1$ segments smoothly concatenated sequentially. $a, b, c \geq 0$ are the power law function coefficients at the initial segment, $\Delta b_1, \ldots, \Delta b_n \in \mathbb{R}$ are the changes in slope in the log-$y$ space, $e_1, \ldots, e_n > 0$ are the location of the breaks to be estimated, and $s_1, \ldots, s_n > 0$ control the smoothness of the transitions between two consecutive segments. Note that the $j$-th segment is an increasing function when $b + \sum_{i=1}^{j-1} \Delta b_i < 0$. The number of breaks $n$ is selected with cross-validation since it is non-differentiable.

**Limitation of the point estimation methods.** Although $\mathcal{M}_4$ (Alabdulmohsin et al., 2022) and BNSL (Caballero et al., 2022) show reasonable prediction performances on a wide set of neural scaling laws, their study also implies that those extrapolations are inherently uncertain. Let the context denote the observations. For instance, when the context part of the curve is flat, it can either imply that the curve is already at the saturating stage of the power law function, or is before the inflection point and will sharply decrease further, following the intuition of $\mathcal{M}_4$. Also, when the context part of the curve is given with its last segment increasing, the BNSL function family cannot reflect the empirical fact that in most cases the curve will eventually start decreasing at some uncertain future point, but will predict that the curve will monotonically increase indefinitely (Fig. 1). Therefore, the natural question is how we can incorporate those uncertainties into the neural scaling law extrapolation to obtain better prediction performance as well as reasonable uncertainties that could be used later for many decision-making problems, *e.g.*, to trade off the cost of collecting and training on more data and the believed associated performance gains.

## 2.2. Prior-data Fitted Networks

In this paper, we propose using Prior-data Fitted Networks (PFNs; Müller et al., 2021), a recently introduced in-context Bayesian prediction method based on the Transformer architecture (Vaswani, 2017). Unlike the conventional Bayesian inference methods that attempt to approximate the posterior distribution of some latent (*e.g.*, the network parameters or the function itself), PFNs directly learn a single transformer that maps a context set to the

*posterior predictive distribution (PPD)*. Learning such a transformer is done with a meta-learning framework. The meta-training data is generated from a *prior distribution* as *synthetic* data, allowing for a potentially infinite supply of examples. The data is then split into a context (training) and a target (test) set, and the transformer model is trained to maximize the likelihood of the target given the context:

$$f \sim p(f), \quad (6)$$

$$(\mathcal{C}, \mathcal{T}) = \mathcal{D} \sim p(\mathcal{D}|f), \quad (7)$$

$$\max_\theta \mathbb{E}_{\mathcal{C}, \mathcal{T} \sim p(\mathcal{D})} \left[ \log q_\theta(Y^*|X^*, \mathcal{C}) \right], \quad (8)$$

where $f$ is a function, $p(f)$ and $p(\mathcal{D}|f)$ are the functional prior and the likelihood for generating the synthetic data $\mathcal{D} = \{(x_i, y_i)\}_{i=1}^{M+N}$, respectively. $\mathcal{D}$ is then randomly split into the context $\mathcal{C} = (X, Y) = \{(x_i, y_i)\}_{i=1}^{M}$ and the target $\mathcal{T} = (X^*, Y^*) = \{(x_i, y_i)\}_{i=M+1}^{M+N}$. $q_\theta$ is the in-context amortized inference machine such as a Transformer parameterized by $\theta$. Note that Eq. (8) is equivalent to minimizing the following expected KL divergence:

$$\min_\theta \mathbb{E}_{p(\mathcal{C}, X^*)} \left[ \text{KL}[p(Y^*|X^*, \mathcal{C}) \| q_\theta(Y^*|X^*, \mathcal{C})] \right], \quad (9)$$

where $p(Y^*|X^*, \mathcal{C}) = \mathbb{E}_{p(f|\mathcal{C})} \left[ p(Y^*|X^*, f) \right]$ is the true posterior predictive distribution (PPD) marginalized over the true functional posterior $p(f|\mathcal{C})$. In this way, the learning objective of PFN in Eq. (8) minimizes the discrepancy between $q_\theta$ and the true PPD, in expectation over the sampling distribution $p(\mathcal{C}, X^*)$. Consequently, after meta-training, given any context $\mathcal{C}$, $q_\theta$ can learn to recover the true PPD at any target points $x^* \in X^*$, as long as a sufficiently large architecture and enough training time are given.

**Flexible prior.** The main strength of PFNs is that we only need to sample the synthetic data from $p(\mathcal{D})$. This implies that the functional prior $p(f)$ does not need to be expressed in a tractable form at all since we do not need to measure its density at both the training and evaluation phases. This property is different from the conventional Bayesian inference methods such as Monte-Carlo Markov Chain (MCMC) and variational inference, allowing us to remove any restrictions on $p(f)$ as long as we can efficiently sample from it.

**PFNs for learning curve extrapolation.** Adriaensen et al. (2023) use PFNs for extrapolating learning curves. Here, $p(f)$ is functional prior implicitly defined over the parameters for describing the learning curves, such as the parameters of the power law function and the coefficients for linearly combining a set of different function families. While their motivation for using PFNs is similar to ours, their prior is not tailored to extrapolating the neural scaling law, making it suboptimal for our purposes, as demonstrated empirically in §4. In the next section, we introduce our novel way of

Table 1: **The template distributions for each scaling law segment.**

| Name | Direction | Function | Prior | | |
|------|-----------|----------|-------|---|---|
| $\mathcal{M}_3$ | Down ($\downarrow$) | $y = a(x^{-1} + d)^b$ | $\log a \sim \mathcal{N}(-1, 0.5)$ | $\log(-b) \sim \mathcal{N}(-2, 1)$ | $\log d \sim \mathcal{N}(0, 1)$ |
| $\mathcal{M}_4$ | Down ($\downarrow$) | $x = g(y) = \left(\frac{y}{a(1-y)^\alpha}\right)^{-\frac{1}{b}}$ | $\log a \sim \mathcal{N}(-1, 0.5)$ | $\log(-b) \sim \mathcal{N}(0, 0.5)$ | $\log \alpha \sim \mathcal{N}(0, 0.5)$ |
| BetaCDF | Up ($\uparrow$) | $y = \texttt{BetaCDF}(x; \beta, \beta)^\gamma$ | $\beta \sim \mathcal{U}(0.5, 1)$ | $\log \gamma \sim \mathcal{N}(0, 0.1)$ | |
| Norm | - | $y \leftarrow y(y_{\max} - y_{\min}) + y_{\min}$ | $y_{\max} \sim \mathcal{U}(0.2, 1.2)$ | $y_{\min} \sim \mathcal{U}(0, y_{\max})$ | |
| Noise | - | $y \leftarrow y + \sigma$ | $\log \sigma \sim \mathcal{N}(-4, 1)$ | | |

(a) **Down**.    (b) **Down-Down**.    (c) **Down-Up-Down**.

Figure 2: **Visualization of 20 neural scaling laws sampled from each function family** before applying the `Noise` function in Table 1.

defining the functional prior suitable for neural scaling law extrapolation.

We defer **more discussion on the related work** to Appendix A due to the space constraint.

## 3. Approach

### 3.1. Neural Scaling Law Prior

**Criteria for the functional prior.** Our functional prior should give high density on the reasonable scaling law behaviors such as power law functions, but also should be flexible enough to cover all the functional forms in the real-world examples. Specifically, we propose the following criteria for the functional prior over the neural scaling laws:

1. We make use of the previously developed functional forms such as $\mathcal{M}_3$ or $\mathcal{M}_4$, but for flexibility we **sample from both of the function families**.

2. Based on the observation of BNSL (Caballero et al., 2022), we allow **breaks to randomly occur** in each scaling law example with some probability, and randomly sample the corresponding segments from the diverse functional families. We highlight that this allows the trained PFN to adapt to various scaling curves with arbitrary breaks, and to *automatically infer* the appropriate number of breaks *without* any validation process.

3. In order to express chaotic behaviors such as double descent (Nakkiran et al., 2021), we let **some segments exhibit upward trends** with some probability, with `BetaCDF`, *i.e.*, the cumulative distribution function of the beta distribution.

**Defining the function families.** We now introduce the template distributions for sampling a complete scaling curve

$\mathcal{D} = \{(x_i, y_i)\}_{i=1}^{M+N}$. We define $\mathcal{M}_3$, $\mathcal{M}_4$, `BetaCDF`, `Norm`, and `Noise` in Table 1; each can be applied to every segment. For the downward segments, we choose only $\mathcal{M}_3$ and $\mathcal{M}_4$, as $\mathcal{M}_1$ and $\mathcal{M}_2$ are subsets of them. We use `BetaCDF` as upward segments because its S-shape fits well to our purpose. We use `Norm` to re-normalize each segment using the sampled maximum and minimum values. We use `Noise` function to apply the observational noise in the likelihood function. We set the parameters $c$ and $y_0$ in Eq. (4) to 0 and 1 for $\mathcal{M}_4$, respectively, as the sampled curves are re-normalized using `Norm`.

Based on those basic template distributions, we define the following function families:

1. **Down**: This function family considers simple downward trends without any breaks, using either $\mathcal{M}_3$ or $\mathcal{M}_4$. Specifically, we randomly choose between $\mathcal{M}_3$ and $\mathcal{M}_4$, sample the parameters, re-normalize each curve with `Norm`, and add noise with `Noise`.

2. **Down-Down**: This function family also represents downward trends but more complex behaviors with breaks. We first randomly choose the number of breaks in $\{1, 2, 3\}$ (resulting in 2, 3, or 4 segments), and randomly sample the locations of those breaks along the $x$-axis. We then randomly select either $\mathcal{M}_3$ or $\mathcal{M}_4$ for each segment, sample the parameters, re-normalize each segment with `Norm`, and add noise with `Noise`. Each segment is translated along the $y$-axis to align with the last $y$ value of the previous segment.

3. **Down-Up-Down**: This function family is to express more complicated behaviors such as double descent (Nakkiran et al., 2021). Here, we fix the number of breaks at 2 (resulting in 3 segments) and randomly sample their locations along the $x$-axis. We use either $\mathcal{M}_3$ or $\mathcal{M}_4$ for the first and third segments, while the

upward function `BetaCDF` is used for the second segment. After sampling the parameters, we re-normalize each segment with `Norm`, add noise with `Noise`, and translate the segments along the $y$-axis to match the last $y$ value of each preceding segment.

Fig. 2 visualizes the curves sampled from the three function families, before applying the `Noise` function. We reject any curves including negative $y$ values or `NaN`.

**Defining the cutoff distribution.** After sampling a complete scaling curve $\mathcal{D}$, we need to decide a cutoff position $M$ to split $\mathcal{D}$ into the context $\mathcal{C} = \{(x_i, y_i)\}_{i=1}^{M}$ and the target $\mathcal{T} = \{(x_i, y_i)\}_{i=M+1}^{M+N}$ for training. However, in defining the cutoff position, we need to think carefully about what kinds of extrapolation problems our model can somehow solve in a reliable manner, and what other types of problems we need to consider as inherently difficult.

Suppose the model is given the context part of the curve, *i.e.*, up to just before the cutoff, and the model needs to predict the rest, *i.e.*, after the cutoff. We can consider the two cases. **1. If the last part of the curve is downward**, we had better assume that a similar tendency will be maintained in the future, *e.g.*, the same power law function. This assumption comes from the intuition (and our empirical observation) that in this case, assuming unexpected future breaks are inherently too difficult to predict due to the lack of information (because almost anything could happen), leading to excessive uncertainties with poor predictive performances. On the other hand, **2. if the last part of the curve is upward**, we already know empirically that the curve will eventually turn downward at some future points, *e.g.*, as we spend more computational resources. Therefore, in this case, we can say that such a downward trend gives the model some useful information about how to predict the future.

Based on those intuitions, we restrict the cutoff position for each function family as follows: **1. Down**: The cutoff position can be at any point. **2. Down-Down**: The cutoff position can only be within the last segment. **3. Down-Up-Down**: The cutoff position can only be at the second or third (last) segments. Note that such restriction does not violate the PFN framework, as it simply specifies $\mathcal{C}, \mathcal{T} \sim p(\mathcal{D})$ in Eq. (8), which defines the procedure of randomly splitting the context and target points. For instance, LC-PFN (Adriaensen et al., 2023) restricts the target points $\mathcal{T}$ to be positioned after the context points $\mathcal{C}$, similarly to ours.

### 3.2. Training Objective

**Context regression loss.** We use a similar objective function to Eq. (8) to train our PFN but add an auto-regressive objective on the context points as follows:

$$\max_{\theta} \mathbb{E}_{\mathcal{C}, \mathcal{T} \sim p(\mathcal{D})} \left[ \log q_\theta(Y^*|X^*, \mathcal{C}) + \log q_\theta(Y|X, \mathcal{C}) \right],$$
(10)

where the second term in the expectation is the auto-regressive objective. We find that adding it improves the curve fit around the cutoff position, both in terms of accuracy and the quality of uncertainty, as shown in Table 6.

**Interpolation loss.** Note that Eq. (10) is primarily for extrapolation (*i.e.*, $\mathcal{T}$ always follows $\mathcal{C}$) and, therefore, lacks the ability to *interpolate* within each curve. On the other hand, in real-world applications such as Bayesian active learning (Gal et al., 2017) (in §4.2), improving the quality of prediction by collecting additional data is often the main interest, but they require the model to be equipped with a good interpolation mechanism. To this end, we also consider training a variant of NSL-PFN that learns both interpolation and extrapolation at the same time. Simply, we first sample the context $\mathcal{C}$ and target $\mathcal{T}$ following §3.1, and then randomly sample a subset of $\mathcal{T}$ to add them to $\mathcal{C}$. We then use the same objective in Eq. (10) for training.

## 4. Experiments

**Datasets.** Following the previous work (Alabdulmohsin et al., 2022; Caballero et al., 2022), we first validate our NSL-PFN on the popular benchmark datasets (Alabdulmohsin et al., 2022) consisting of scaling curves evaluated on various tasks in both image and natural language domain. The image classification (**IC**) dataset includes 72 scaling curves, each of which evaluates few-shot prediction performances of various neural network architectures w.r.t. the number of training datapoints. Specifically, many popular architectures such as BiT (Kolesnikov et al., 2020), MiX (Tolstikhin et al., 2021), and ViT (Alexey, 2020) are evaluated on the various image classification datasets such as ImageNet (Russakovsky et al., 2015), CIFAR100 (Krizhevsky et al., 2009), Birds (Welinder et al., 2010), and Caltech101 (Fei-Fei et al., 2004). The natural language processing (**NLP**) dataset contains 20 scaling curves, each of which evaluates the performance of various Transformer architectures (Bansal et al., 2022; Thoppilan et al., 2022) w.r.t. the number of training datapoints, on neural machine translation (NMT), language modeling (LM), and Big-Bench (BB; bench authors, 2023). We further consider the nanoGPT-Bench dataset (**Nano**; Kadra et al., 2023) consisting of 24 scaling curves obtained by training nanoGPT models with varying model sizes on the OpenWebText dataset (Gokaslan et al., 2019). We also use **ColPret**, a recently released huge dataset containing more than 1000 curves (Choshen et al., 2024), each of which evaluates the performance various LLMs w.r.t. the number of training datapoints. We subsample it into 192 curves such that the number of points in each curve lies within $[10, 1000]$. Lastly, we consider double descent (**DD**) dataset (Nakkiran et al., 2021) consisting of 16 curves exhibiting double descent behavior, to test each model's ability to predict more complex scaling behaviors. See Appendix B for details.

Table 2: **Results on image domain tasks (IC)**. We report mean and standard deviation (std) over 3 runs for the MCMC baselines and our method. For the other methods, we do not report std, as the point estimation methods always produce the same results, and for LC-PFN, we use the single model released by the authors. The best results are in **bold**, and the second-best results are underlined.

| Method | Bayes. | ImageNet | | CIFAR100 | | Birds | | Caltech101 | | Average | |
|---|---|---|---|---|---|---|---|---|---|---|---|
| | | RMSLE ($\downarrow$) | LL ($\uparrow$) | RMSLE ($\downarrow$) | LL ($\uparrow$) | RMSLE ($\downarrow$) | LL ($\uparrow$) | RMSLE ($\downarrow$) | LL ($\uparrow$) | RMSLE ($\downarrow$) | LL ($\uparrow$) |
| $\mathcal{M}_1$ | ✗ | 0.0838 | - | 0.0803 | - | 0.0749 | - | 0.1793 | - | 0.1046 | - |
| $\mathcal{M}_2$ | ✗ | 0.0830 | - | 0.0626 | - | 0.0729 | - | 0.1105 | - | 0.0823 | - |
| $\mathcal{M}_3$ | ✗ | 0.0470 | - | 0.0436 | - | 0.0555 | - | 0.1000 | - | 0.0615 | - |
| $\mathcal{M}_4$ | ✗ | 0.0158 | - | **0.0315** | - | 0.0302 | - | 0.0886 | - | 0.0415 | - |
| BNSL | ✗ | 0.0121 | - | 0.0351 | - | 0.0242 | - | 0.0911 | - | 0.0406 | - |
| MCMC ($\mathcal{M}_1$) | ✓ | 0.0794±0.0011 | 1.934±0.026 | 0.0812±0.0010 | 1.672±0.040 | 0.0724±0.0048 | 1.993±0.036 | 0.1802±0.0057 | 1.957±0.022 | 0.1047±0.0010 | 1.930±0.020 |
| MCMC ($\mathcal{M}_2$) | ✓ | 0.0789±0.0040 | 1.974±0.041 | 0.0618±0.0001 | 2.344±0.012 | 0.0662±0.0018 | 2.191±0.008 | 0.1056±0.0013 | 2.582±0.029 | 0.0776±0.0012 | 2.287±0.013 |
| MCMC ($\mathcal{M}_3$) | ✓ | 0.0421±0.0025 | 2.655±0.023 | 0.0424±0.0004 | 2.663±0.006 | 0.0572±0.0017 | 2.348±0.025 | 0.0978±0.0009 | 2.627±0.020 | 0.0595±0.0005 | 2.568±0.004 |
| MCMC ($\mathcal{M}_4$) | ✓ | 0.0160±0.0006 | 3.454±0.073 | 0.0320±0.0008 | **2.969±0.033** | 0.0316±0.0015 | 3.039±0.028 | 0.0811±0.0022 | 2.733±0.002 | 0.0422±0.0006 | 3.024±0.016 |
| MCMC (BNSL) | ✓ | 0.0109±0.0002 | 3.562±0.042 | 0.0412±0.0002 | 2.649±0.018 | 0.0277±0.0032 | 3.149±0.032 | 0.1782±0.1023 | 2.324±0.008 | 0.0645±0.0253 | 2.921±0.002 |
| LC-PFN | ✓ | 0.0096 | **4.146** | 0.0373 | 1.644 | 0.0233 | 2.861 | 0.1009 | 1.065 | 0.0428 | 2.429 |
| **NSL-PFN (ours)** | ✓ | **0.0092±0.0009** | 3.808±0.0172 | 0.0317±0.0005 | 2.829±0.0426 | **0.0202±0.0006** | **3.464±0.0419** | **0.0507±0.0032** | **3.217±0.0472** | **0.0280±0.0009** | **3.330±0.0136** |

Table 3: **Results on language domain tasks (NLP and Nano)**.

| Method | Bayes. | NMT | | LM | | BB | | Nano | | Average | |
|---|---|---|---|---|---|---|---|---|---|---|---|
| | | RMSLE ($\downarrow$) | LL ($\uparrow$) | RMSLE ($\downarrow$) | LL ($\uparrow$) | RMSLE ($\downarrow$) | LL ($\uparrow$) | RMSLE ($\downarrow$) | LL ($\uparrow$) | RMSLE ($\downarrow$) | LL ($\uparrow$) |
| $\mathcal{M}_1$ | ✗ | 0.2217 | - | 0.0140 | - | 0.0148 | - | 0.0534 | - | 0.0593 | - |
| $\mathcal{M}_2$ | ✗ | 0.0341 | - | 0.0010 | - | 0.0146 | - | 0.0549 | - | 0.0367 | - |
| $\mathcal{M}_3$ | ✗ | 0.0471 | - | 0.0022 | - | 0.0087 | - | 0.0534 | - | 0.0367 | - |
| $\mathcal{M}_4$ | ✗ | 0.0208 | - | **0.0009** | - | 0.0123 | - | 0.0360 | - | 0.0249 | - |
| BNSL | ✗ | 0.0184 | - | 0.0016 | - | 0.0164 | - | 0.0299 | - | 0.0223 | - |
| MCMC ($\mathcal{M}_1$) | ✓ | 0.2247±0.0055 | 0.036±0.240 | 0.0155±0.0008 | 2.746±0.088 | 0.0158±0.0006 | 2.958±0.054 | 0.0515±0.0008 | 1.232±0.102 | 0.0569±0.0006 | 1.732±0.020 |
| MCMC ($\mathcal{M}_2$) | ✓ | 0.0333±0.0015 | 2.791±0.020 | 0.0023±0.0006 | 3.695±0.046 | 0.0143±0.0006 | 2.974±0.057 | 0.0521±0.0012 | 0.352±0.387 | 0.0365±0.0007 | 1.636±0.219 |
| MCMC ($\mathcal{M}_3$) | ✓ | 0.0471±0.0000 | 2.592±0.000 | 0.0023±0.0000 | 3.072±0.000 | 0.0086±0.0007 | 3.065±0.035 | 0.0537±0.0005 | 0.411±0.226 | 0.0368±0.0004 | 1.716±0.119 |
| MCMC ($\mathcal{M}_4$) | ✓ | **0.0144±0.0017** | 3.090±0.053 | 0.0035±0.0005 | 3.331±0.047 | **0.0083±0.0004** | 3.098±0.036 | 0.0374±0.0006 | 1.474±0.060 | 0.0235±0.0004 | 2.216±0.038 |
| MCMC (BNSL) | ✓ | 0.0438±0.0207 | 2.842±0.072 | 0.0022±0.0002 | **3.969±0.064** | 0.0201±0.0006 | 2.747±0.035 | 0.0887±0.0282 | 2.076±0.037 | 0.0582±0.0134 | 2.531±0.007 |
| LC-PFN | ✓ | 0.0471 | -0.354 | 0.1151 | 2.805 | 0.0191 | 1.019 | 0.0311 | 1.849 | 0.0398 | 1.519 |
| **NSL-PFN (ours)** | ✓ | 0.0248±0.0102 | 2.798±0.0916 | 0.0013±0.0004 | 3.740±0.0059 | 0.0202±0.0006 | **3.321±0.0391** | **0.0260±0.0027** | **2.339±0.1029** | **0.0194±0.0021** | **2.773±0.0605** |

Table 4: **Results on ColPret and double descent (DD) dataset.**
† indicates that a few trials failed due to overflow error, which were excluded from the calculation.

| Method | ColPret | | DD | |
|---|---|---|---|---|
| | RMSLE ($\downarrow$) | LL ($\uparrow$) | RMSLE ($\downarrow$) | LL ($\uparrow$) |
| $\mathcal{M}_1$ | 0.0732 | - | 0.1140 | - |
| $\mathcal{M}_2$ | 0.0545† | - | 0.1140 | - |
| $\mathcal{M}_3$ | 0.0837 | - | 0.1333 | - |
| $\mathcal{M}_4$ | 0.0410 | - | 0.0910 | - |
| BNSL | 0.0998 | - | 0.0468 | - |
| MCMC ($\mathcal{M}_1$) | 0.0769±0.0022 | 1.334±0.062 | 0.1206±0.0054 | -10.561±0.181 |
| MCMC ($\mathcal{M}_2$) | 0.0533±0.0010 | 1.835±0.033 | 0.1199±0.0109 | -10.842±0.271 |
| MCMC ($\mathcal{M}_3$) | 0.0504±0.0031 | 1.798±0.017 | 0.1222±0.0093 | -2.782±2.142 |
| MCMC ($\mathcal{M}_4$) | 0.0417±0.0302 | 1.787±0.120 | 0.0925±0.0061 | -0.255±0.606 |
| MCMC (BNSL) | 0.0690±0.0127 | 2.727±0.011 | 0.0494±0.0017 | 1.250±0.244 |
| LC-PFN | 0.0289 | **2.804** | 0.0706 | 1.321 |
| **NSL-PFN (ours)** | **0.0271±0.0003** | 2.794±0.025 | **0.0335±0.0013** | **2.565±0.023** |

**Baselines.** We first consider point estimation methods, including $\mathcal{M}_{1-4}$ and **BNSL**, as described in §2.1. For Bayesian baselines, we compare against a **MCMC** method—specifically, EMCEE (Foreman-Mackey et al., 2013)—following the setup of Domhan et al. (2015). We evaluate five variants of EMCEE, each corresponding to the MCMC version of one of the function families: $\mathcal{M}_{1-4}$ and BNSL. The likelihood function is defined as $\mathcal{N}(y; f(x), \sigma^2)$, where $\sigma^2$ is a parameter capturing observational noise and $f$ is one of the models from $\mathcal{M}_{1-4}$ or BNSL. The number of MCMC samples (nsamples) is set to 150 (except in Fig. 6) to ensure that the inference time is comparable to that of NSL-PFN. We also compare against **LC-PFN** (Adriaensen et al., 2023), another Bayesian method originally developed for extrapolating learning curves. Since LC-PFN assumes monotonically increasing functions, we horizontally flip the model output, i.e., $\hat{y} \leftarrow 1 - \hat{y}$. See Appendix C for further details on the baselines, including prior settings for the MCMC variants and other hyperparameter configurations used with EMCEE.

**Evaluation metric.** We report the root mean squared log error (**RMSLE**; Alabdulmohsin et al., 2022; Caballero et al., 2022), i.e., $\sqrt{\frac{1}{|\mathcal{T}|} \sum_{y^* \in \mathcal{T}} (\log \hat{y} - \log y^*)^2}$, where $\mathcal{T}$ is the target set, $\hat{y}$ the prediction[1], and $y^*$ the corresponding label. For the Bayesian methods, we also evaluate the quality of predictive uncertainties by reporting the average log-likelihood (**LL**), i.e., $\frac{1}{|\mathcal{T}|} \sum_{(x^*, y^*) \in \mathcal{T}} \log q(y^* | x^*, \mathcal{C})$, where $q$ is the predictive distribution of each method, and $\mathcal{C}$ and $\mathcal{T}$ are the context and target set, respectively. LL and RMSLE are averaged across all the curves in each dataset.

**Implementation.** We mostly follow Müller et al. (2021). We use a Transformer (Vaswani, 2017) and treat each context pair $(x, y)$ and target $x^*$ as individual tokens while omitting positional encoding. We discretize the output distribution into a finite number of bins (1,000). The hyperparameters of architecture are set as follows: nlayers=12, nheads=4, and nhidden=512. We train our model on 1.6M synthetic examples sampled from our prior for 100K iterations. See Appendix C for more details.

---

[1] We use mean for the MCMC baselines, and the median for both LC-PFN and NSL-PFN, following Adriaensen et al. (2023).

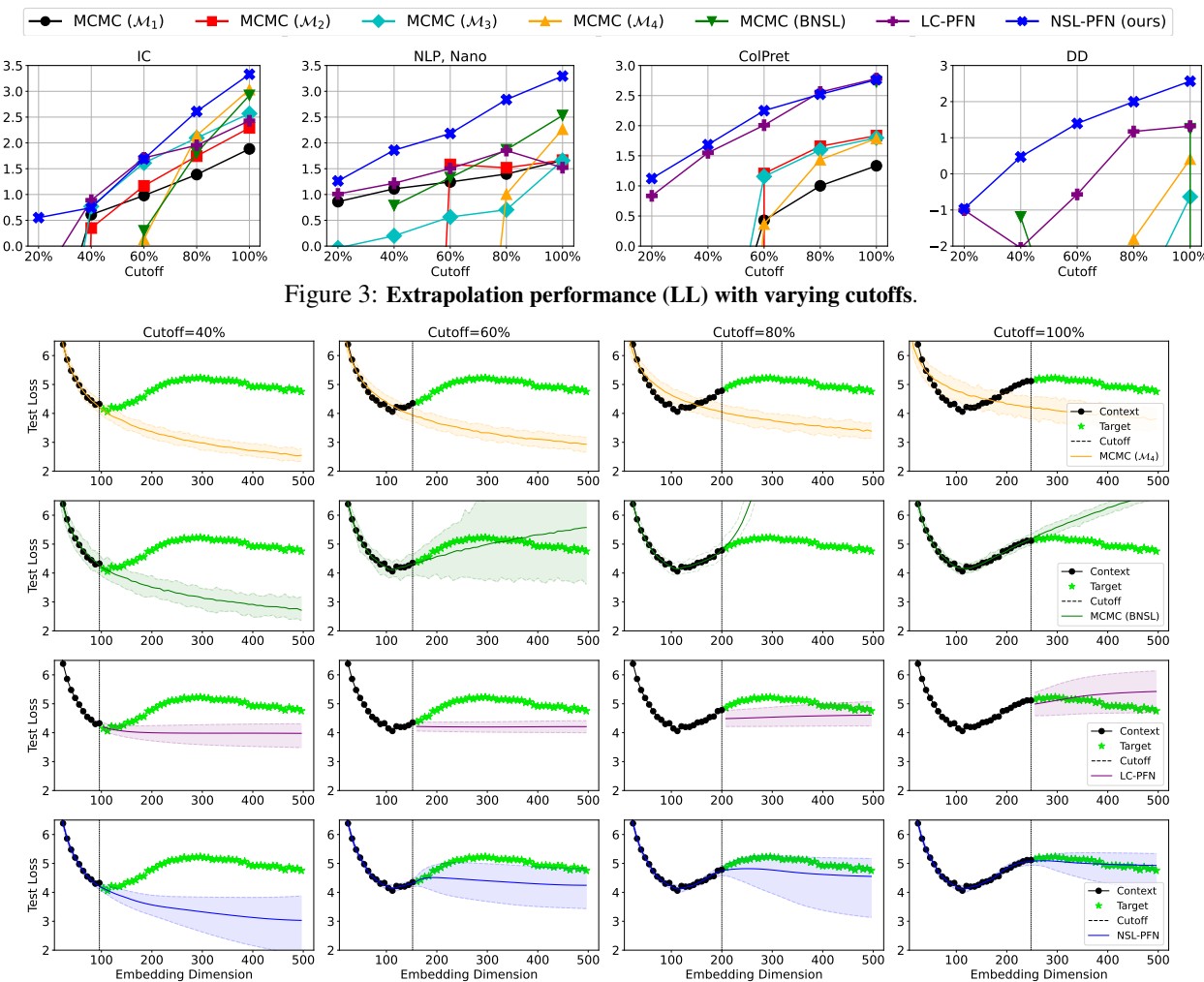

Figure 3: **Extrapolation performance (LL) with varying cutoffs.**

Figure 4: **Visualization of extrapolation on double descent (DD) (Nakkiran et al., 2021) with varying cutoffs.** We visualize the mean or median as solid lines and show the 90% confidence intervals with shaded areas. All the visualizations are presented in Appendix E.

## 4.1. Main Experimental Results

**Quantitative analysis.** We first evaluate our NSL-PFN on the popular benchmark datasets on both image (IC, in Table 2) and language domain (NLP and Nano, in Table 3). On both domains, our NSL-PFN consistently outperforms the baselines in general, achieving the best RMSLE and LL on average (on the rightmost column for each table). Whereas NSL-PFN significantly performs better than the baselines on almost all the cases, it performs slightly worse than MCMC ($\mathcal{M}_4$) and $\mathcal{M}_4$ on NMT and LM, respectively. We conjecture that this is because the scaling laws on those cases are rather too simple (see Fig. 12 and Fig. 13 for the visualization) such that considering complicated cases such as double descent in our prior design may act as a distractor and thus lead to worse predictive performance. Except for such few cases, overall, the results clearly show the superiority of our NSL-PFN in terms of both predictive accuracy and the quality of uncertainty. We also test NSL-PFN on the recently released huge dataset called ColPret,

and the DD dataset in Table 4. The results clearly show that our NSL-PFN performs better than all the baselines in terms of both metrics, especially more significantly on DD, demonstrating the strong ability of NSL-PFN to predict complex scaling behaviors in a reliable manner.

**Effect of context set size.** In Fig. 3, we further test the robustness in prediction against varying context set size, *i.e.*, the number of observations in each curve, or the cut-off. We consider only the Bayesian methods since it is Bayesian that aims to make models more robust against fewer observations, such as how well the predictive uncertainty (coming from the limited observations) covers the unseen target points. We thus use LL to evaluate both the predictive accuracy and quality of uncertainty at the same time. We can see that our NSL-PFN shows higher LL on all the datasets and various cutoff percentages, clearly showing its robustness in prediction over the varying sizes of context sets. The gap is especially prominent on DD as expected, primarily because our functional prior already covers di-

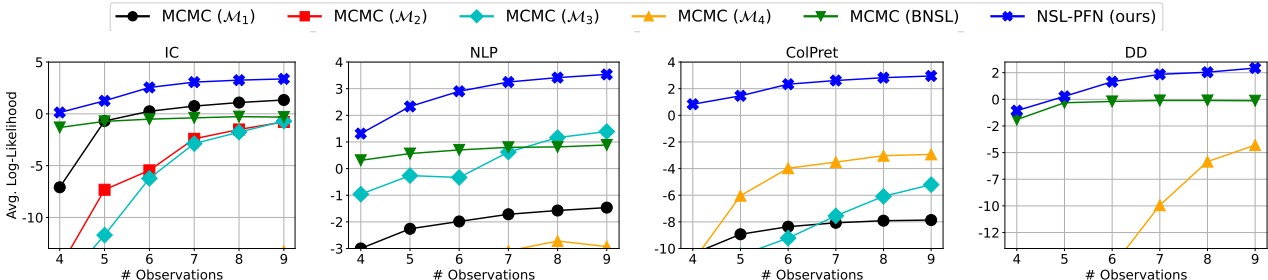

Figure 5: **Results of Bayesian active learning experiments.**

verse forms of double descent behavior with varying cutoffs, with the resultant posterior predictive distribution (PPD) being more robust against such variation and data scarcity.

**Qualitative analysis.** In Fig. 4, we visualize the extrapolation on DD to better understand how each model actually predicts with varying cutoffs. MCMC ($\mathcal{M}_4$) struggles whenever the context part includes the upward segment (*e.g.*, the upper row), because its base function family $\mathcal{M}_4$ cannot express such non-monotonic behavior by definition. MCMC (BNSL) is able to express such upward trends but struggles to generalize whenever the last segment of the context part is increasing, predicting that the curve will increase indefinitely (*e.g.*, the last two panels). On the other hand, NSL-PFN shows reasonable predictions even in such cases, predicting that the curve will eventually decrease at some uncertain future points, as intended. The leftmost panel can be seen as the only failure case. As explained in §3.1, we consider such a case as inherently difficult to correctly predict, assuming that there is no way to infer from the single decreasing segment that the curve will go upward. Lastly, LC-PFN shows suboptimal performance than NSL-PFN because its functional prior is unaware of such complex behavior. See Appendix E for other visualizations on DD.

## 4.2. Bayesian Active Learning

Considering that scaling curves are often collected from training large neural networks with huge amounts of data, collecting each observation point can be very costly. Therefore, we must carefully decide *where to observe* to minimize the cost, especially in real-world scenarios involving important decision-making problems such as how much cost should be spent to achieve the desired performance. We frame this as a Bayesian active learning (BAL; Houlsby et al., 2011; Gal et al., 2017) problem—incrementally observing the unseen point that is expected to maximize the average likelihood on all the other unseen points, based on the uncertainty information provided by the given model.

In this experiment, starting from four observations, we iteratively select the next unseen point with a specific criterion. We use variation ratio, *i.e.*, $\max_x [1 - \max_y q(y|x, \mathcal{C})]$ (Freeman, 1965) as our acquisition function, which prioritizes the unseen area where the given model is least confident. For NSL-PFN, we use the model trained to predict interpo-

Table 5: **Ablation study on the prior distribution**.

| Prior | Break | Up | IC | | NLP, Nano | | ColPret | | DD | |
|---|---|---|---|---|---|---|---|---|---|---|
| | | | RMSLE | LL | RMSLE | LL | RMSLE | LL | RMSLE | LL |
| $\mathcal{M}_3$ | ✗ | ✗ | 0.064 | 2.54 | 0.026 | 1.57 | 0.035 | 1.94 | 0.131 | -1.71 |
| $\mathcal{M}_4$ | ✗ | ✗ | 0.050 | 2.90 | 0.027 | 2.83 | 0.075 | 2.24 | 0.058 | 2.01 |
| $\mathcal{M}_{3,4}$ | ✗ | ✗ | 0.038 | 2.90 | **0.015** | **2.92** | 0.032 | 2.23 | 0.071 | 1.20 |
| $\mathcal{M}_{3,4}$ | ✓ | ✗ | 0.032 | 3.24 | 0.015 | 2.82 | 0.027 | 2.74 | 0.051 | 2.12 |
| $\mathcal{M}_{3,4}$ | ✓ | ✓ | **0.028** | **3.33** | 0.019 | 2.77 | **0.027** | **2.79** | **0.033** | **2.57** |

Table 6: **Ablation study on the context regression loss**.

| Context loss | IC | | NLP, Nano | | ColPret | | DD | |
|---|---|---|---|---|---|---|---|---|
| | RMSLE | LL | RMSLE | LL | RMSLE | LL | RMSLE | LL |
| ✗ | 0.0309 | 3.236 | 0.0205 | 2.630 | **0.0263** | 2.769 | 0.0322 | 2.377 |
| ✓ | **0.0280** | **3.330** | **0.0194** | **2.773** | 0.0271 | **2.794** | 0.0335 | **2.565** |

lations between context points, as explained in §3.2. Note that LC-PFN cannot be used in this experiment since it is not learned to interpolate within each curve.

Fig. 5 shows the results on all the datasets except for Nano, since each curve in Nano contains only 7 points in total, which is too few for conducting this experiment. We can see that the overall performance of NSL-PFN is much better than the baselines, and also consistently improves as more observations are collected. The results clearly demonstrate the superiority of our model in terms of the quality of uncertainty, positioning NSL-PFN as a valuable tool that can help make many important decisions in the real world.

## 4.3. Other Analysis

**Ablation study.** We validate criteria used for defining our functional prior explained in §3.1. In Table 5, we see that using both $\mathcal{M}_3$ and $\mathcal{M}_4$ is better than using only one of them in general (1st and 2nd row vs. 3rd row). The assumption of random breaks also improves the performance (3rd vs. 4th row). Lastly, including the upward trend improves the performance slightly on IC and ColPret, largely on DD as expected, but not on NLP and Nano (4th vs. last row). This is because, as mentioned in §4.1, the scaling laws in the NLP dataset are too simple and do not require training with such unexpected behavior. We further examine in Table 6 the effectiveness of the context regression loss in Eq. (10). In general, including it improves both metrics, except for a slight decrease of RMSLE on the ColPret and DD datasets.

**Efficiency.** We highlight the efficiency of amortized inference compared to non-amortized baselines such as point estimation and MCMC. As shown in Table 7, the inference time for both point estimation and MCMC are significantly

Table 7: **Avg. inference time (s) per curve (nsamples=150).**

| $\mathcal{M}_4$ | BNSL | MCMC ($\mathcal{M}_4$) | MCMC (BNSL) | LC-PFN | NSL-PFN (ours) |
|---|---|---|---|---|---|
| 15.65 | 98.79 | 154.64 | 280.55 | **0.02** | **0.02** |

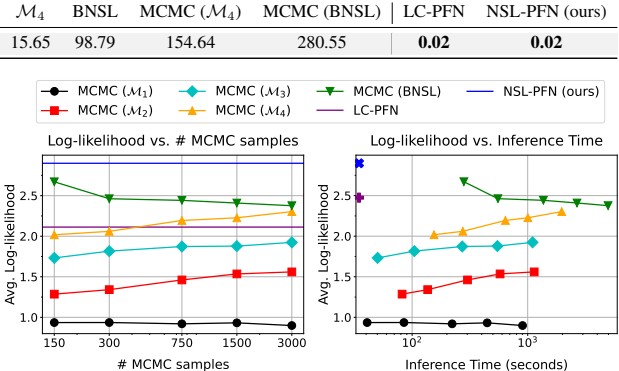

Figure 6: **MCMC with an increased number of samples (nsamples).** Convergence failures were observed for some MCMC (BNSL) runs, and these runs are excluded.

higher than those of amortized approaches like LC-PFN and our NSL-PFN. This efficiency arises because, once meta-trained, PFN-based methods require only a single forward pass per curve to make a prediction. BNSL and its MCMC variant are particularly slow due to an additional validation process for selecting the number of breakpoints.

Furthermore, we investigate the effect of increasing the number of MCMC samples (nsamples = 300, 750, 1500, 3000) on performance. As illustrated in Fig. 6, these experiments show that increasing the number of samples yields only marginal performance gains, which are not commensurate with the additional computational cost. In fact, for MCMC ($\mathcal{M}_1$) and MCMC (BNSL), performance slightly degrades with longer sampling chains. Crucially, all baselines still underperform compared to NSL-PFN, showing both its effectiveness and efficiency.

**BLR and DKGP.** Beyond MCMC-based approaches, we also explore other Bayesian inference baselines for extensive comparison. Specifically, we include Bayesian Linear Regression (**BLR**; Bishop, 2006) with neural network (NN) basis functions and Deep Kernel Gaussian Process (**DKGP**; Wilson et al., 2016). For BLR with NN, parameters of the neural network serving as basis functions, along with other model parameters, are tuned by maximizing the log marginal likelihood. While these methods—BLR with appropriate basis functions offering interpretability and DKGP providing flexible non-parametric modeling—are standard for 1D regression, they often encounter difficulties in effectively capturing the distinct inductive biases inherent in neural scaling laws, as demonstrated in Table 8. This challenge in embedding specific prior knowledge about scaling behaviors into their respective basis or kernel functions can render them less competitive for extrapolating neural scaling laws when compared to models explicitly designed for this task. Further details regarding the implementation of BLR and DKGP, as well as a discussion of other BLR variants utilizing polynomial, RBF, Fourier, sigmoid, and spline

Table 8: **Results of BLR and DKGP.** ‡ indicates that some RMSLEs could not be calculated due to negative y-values.

| Method | IC | | NLP, Nano | | ColPret | | DD | |
|---|---|---|---|---|---|---|---|---|
| | RMSLE | LL | RMSLE | LL | RMSLE | LL | RMSLE | LL |
| BLR (with NN) | 0.307 | -0.38 | 0.074 | -3.08 | 0.507 | -2.73 | 0.603 | -95.97 |
| DKGP | 0.197‡ | -1.86 | 0.067 | 0.62 | 0.101‡ | 0.10 | 0.143 | -3.29 |
| **NSL-PFN (ours)** | **0.028** | **3.33** | **0.019** | **2.77** | **0.027** | **2.79** | **0.033** | **2.57** |

Table 9: **Results of calibration using MSCE (↓).**

| Method | IC | NLP, Nano | ColPret | DD |
|---|---|---|---|---|
| MCMC ($\mathcal{M}_1$) | $0.1422_{\pm0.1365}$ | $0.2328_{\pm0.0239}$ | $0.1349_{\pm0.0035}$ | $0.1676_{\pm0.0315}$ |
| MCMC ($\mathcal{M}_2$) | $0.0912_{\pm0.0046}$ | $0.1922_{\pm0.0076}$ | $0.1096_{\pm0.0047}$ | $0.2394_{\pm0.0289}$ |
| MCMC ($\mathcal{M}_3$) | $0.0538_{\pm0.0624}$ | $0.2048_{\pm0.0070}$ | $0.1010_{\pm0.0006}$ | $0.2641_{\pm0.0171}$ |
| MCMC ($\mathcal{M}_4$) | $\mathbf{0.0381_{\pm0.0534}}$ | $0.1630_{\pm0.0072}$ | $0.1169_{\pm0.0057}$ | $0.1943_{\pm0.0078}$ |
| MCMC (BNSL) | $0.0773_{\pm0.0016}$ | $0.1238_{\pm0.0048}$ | $0.0994_{\pm0.0056}$ | $0.2072_{\pm0.0094}$ |
| LC-PFN | 0.1809 | 0.2674 | 0.1071 | $\underline{0.1500}$ |
| **NSL-PFN (ours)** | $\underline{0.0512_{\pm0.0014}}$ | $\mathbf{0.1145_{\pm0.0271}}$ | $\mathbf{0.0615_{\pm0.0053}}$ | $\mathbf{0.0826_{\pm0.0012}}$ |

bases, are available in Appendix C.

**Calibration.** To additionally assess the quality of predictive uncertainties, we incorporate the mean square calibration error (**MSCE**; Kuleshov et al., 2018), i.e., $\frac{1}{J}\sum_{j=1}^{J} w_j \cdot (p_j - \hat{p}_j)^2$, where $m$ is the number of bins, $p_j$ represents the predicted confidence level, $\hat{p}_j$ is the empirical frequency of the true outcome falling within that confidence level, and $w_j$ are weights (typically set to 1). This metric evaluates how well the predicted confidence levels align with the actual observed frequencies. As shown in Table 9, NSL-PFN demonstrates notably better calibration performance than baseline models, particularly on ColPret and DD. These results show the robustness of NSL-PFN in handling realistic scaling behaviors and the importance of well-calibrated uncertainty for reliable extrapolation in such regimes.

**Prior hyperparameter tuning.** For our prior design, we manually adjust the parameters of our functional prior to visually match the shapes of the actual curves collected from various domains. To assess whether NSL-PFN can further improve prior hyperparameter tuning, we performed Bayesian optimization on the prior parameters (e.g., $a, b, d$ for $\mathcal{M}_3$ and $a, b, \alpha$ for $\mathcal{M}_4$). The optimization process successfully identified configurations that yielded slight performance improvements over our default settings with just 60 BO steps. See Appendix D for details.

## 5. Conclusion

In this work, we introduced a Bayesian approach using Prior-data Fitted Networks (PFNs) to improve neural scaling law extrapolation, addressing limitations of traditional point estimation by incorporating uncertainty quantification. By designing a prior distribution that enables PFNs to meta-learn through synthetic functions resembling real-world scaling laws, our method effectively extrapolates scaling behavior in diverse scenarios. Evaluation on real-world scaling laws shows superior performance, particularly in data-limited settings like Bayesian active learning, demonstrating its usefulness for practical and uncertainty-sensitive applications.

## Acknowledgements

This work was partially supported by Institute of Information & communications Technology Planning & Evaluation (IITP) grant funded by the Korea government (MSIT) (No. RS-2024-00457882, AI Research Hub Project), IITP grant funded by MSIT (No. RS-2020-II200153, Penetration Security Testing of ML Model Vulnerabilities and Defense), IITP grant funded by MSIT (No. RS-2019-II190075, Artificial Intelligence Graduate School Program (KAIST)), IITP grant funded by MSIT (No.RS-2022-II220713, Meta-learning Applicable to Real-world Problems), Samsung Electronics (IO201214-08145-01), and National Research Foundation of Korea (NRF) grant funded by MSIT (No. RS-2023-00256259).

## Impact Statement

This paper introduces a Bayesian approach to neural scaling law extrapolation using Prior-data Fitted Networks (PFNs), emphasizing resource-efficient decision-making. By designing a tailored prior for PFNs, our method enables uncertainty-aware predictions, improving over existing point estimation and Bayesian techniques. This enhances the predictability of neural scaling laws, leading to more efficient model training and optimized resource allocation. Reliable extrapolations with quantified uncertainty support better decision-making in large-scale model training, reducing unnecessary computational costs. While our approach improves efficiency, practitioners should complement it with empirical validation to ensure responsible and informed scaling decisions.

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

## A. More Discussion on Related Work

**Neural scaling laws**   There are many observations that neural scaling laws are empirically predictable (Hestness et al., 2017; Johnson et al., 2018; Rosenfeld et al., 2019; Kaplan et al., 2020; Rosenfeld, 2021; Ghorbani et al., 2021; Bansal et al., 2022; Hoffmann et al., 2022; Chung et al., 2024), just to name a few. We only discuss recent studies that have explored parameterizations for neural scaling law extrapolation. The simplest model $\mathcal{M}_1$ assumes a power law function (*e.g.*, $y = ax^{-b}$) throughout the domain of interest $x$ (*e.g.*, data, model size, or training time), where it has been used in various domains, such as healthcase (Cho et al., 2015), neural machine translation (NMT; Gordon et al., 2021), and language models (Kaplan et al., 2020), among others (Hestness et al., 2017; Johnson et al., 2018; Sharma & Kaplan, 2022). The most widely used model in the literature is $\mathcal{M}_2$, which adds a bias term to the power law function, to capture saturating performance for large $x$ (Mukherjee et al., 2003; Figueroa et al., 2012; Domhan et al., 2015; Hestness et al., 2017; Johnson et al., 2018; Rosenfeld et al., 2019; Abnar et al., 2021; Gordon et al., 2021; Rosenfeld, 2021). A different parameterization $\mathcal{M}_3$ has been recently used in NMT (Bansal et al., 2022) to guarantee that the power law function converges to a finite constant when the domain of interest approaches to infinity. Alabdulmohsin et al. (2022) have proposed $\mathcal{M}_4$ to handle non-power law behaviour in partial scaling law. More recently, Caballero et al. (2022) have proposed broken neural scaling law (BNSL) by assuming that real-world neural scaling laws have different power law for each segment (Nakkiran et al., 2021).

**Prior-data Fitted Networks (PFNs)**   In-Context Learning (ICL) represents a paradigm shift from traditional learning approaches by enabling rapid adaptation to new tasks and data without extensive retraining, through conditioning on input prompts or context (Dong et al., 2022). The rise of ICL parallels the development of Transformer-based architectures, particularly large language models (LLMs; Radford et al., 2019; Brown, 2020; Achiam et al., 2023; Touvron et al., 2023a;b; Dubey et al., 2024). Prior-data Fitted Networks (PFNs) are Transformer models designed for in-context Bayesian prediction, facilitating efficient inference in a single forward pass without extensive retraining or hyperparameter tuning. PFNs have demonstrated effectiveness in various applications, such as Bayesian learning curve extrapolation (Adriaensen et al., 2023), in-context tabular classification (Hollmann et al., 2022), black-box hyperparameter optimization (HPO; Müller et al., 2023), freeze-thaw Bayesian optimization (Rakotoarison et al., 2024), and time-series forecasting (Dooley et al., 2024; Verdenius et al., 2024). Building upon this foundation, our work applies PFNs to infer neural scaling laws.

**Learning curve extrapolation**   Learning curves have long been an active area of research, as surveyed by Mohr & van Rijn (2022). Early works (Cortes et al., 1993; Frey & Fisher, 1999; Kolachina et al., 2012) primarily modeled learning curves (LCs) based on dataset size, providing "point estimates" of performance without a probabilistic nature, let alone a Bayesian framework for uncertainty quantification. Additionally, these models often do not address deep neural network (DNN) training, where the x-axis typically represents training time, measured in parameter updates or epochs. Non-Bayesian probabilistic models for DNN training curves were later explored (Chandrashekaran & Lane, 2017; Gargiani et al., 2019). In contrast, the first Bayesian approach for DNN training curve extrapolation was introduced by Domhan et al. (2015) using Markov Chain Monte Carlo (MCMC) methods. More recently, Adriaensen et al. (2023) introduced a method employing Prior-data Fitted Networks (PFNs) to predict LCs based on training time. While this method bears a resemblance to our approach, it is constrained in its applicability to certain scaling behaviors, such as power-law scaling and double descent, seen when the $x$-axis reflects model scale or dataset size. Furthermore, it lacks the flexibility to model variable scaling behaviors across different regions of the $x$-axis, motivating our development of a tailored prior capable of accommodating multiple breaks and facilitating Bayesian inference with user-defined configurations. Moreover, while extrapolation has been investigated in AutoML contexts (Swersky et al., 2014; Klein et al., 2022; Wistuba et al., 2022; Kadra et al., 2023; Rakotoarison et al., 2024; Lee et al., 2024), and future applications of our work may include enabling AutoML for large-scale deep learning, we currently prioritize the enhancement of neural scaling law extrapolation via Bayesian methods, rather than fully automating scaling up the hyperparameter optimization process.

## B. Datasets

**IC.**   We use the scaling laws benchmark (Alabdulmohsin et al., 2022), which covers various scaling laws in both vision and NLP domains. In the vision domain of the benchmark, the downstream evaluation focuses on 5-shot, 10-shot, and 25-shot image classification (**IC**) tasks. It includes two sizes of big-transfer residual neural networks (BiT; Kolesnikov et al., 2020), MLP mixers (MiX; Tolstikhin et al., 2021), and vision transformers (ViT; Alexey, 2020). These models are pretrained on the JFT-300M (Sun et al., 2017) dataset and tested on downstream tasks: **ImageNet** (Russakovsky et al.,

2015), **CIFAR-100** (Krizhevsky et al., 2009), **Birds** (Welinder et al., 2010), and **Caltech101** (Fei-Fei et al., 2004). For each downstream task, the benchmark provides 18 scaling law variations (2 model sizes × 3 model types × 3 few-shot settings), resulting in a total of 72 scaling laws. The x-axis of the scaling law corresponds to the number of examples used for training, while the y-axis represents the error rate in image classification.

**NLP.** In the NLP domain of the benchmark, scaling laws are provided for downstream tasks including neural machine translation (**NMT**), language modeling (**LM**), and Big-Bench (**BB**; bench authors, 2023). For the NMT task, scaling laws are derived from five encoder-decoder configuration combinations trained following the methodology outlined in (Bansal et al., 2022). In this case, the x-axis represents the number of observed examples, while the y-axis corresponds to the log perplexity. For the LM task, the performance of LaMDA architecture (Thoppilan et al., 2022) is evaluated on next-token prediction. Scaling laws are provided for five different parameter sizes, with the x-axis indicating the number of observed examples and the y-axis showing the validation loss, rescaled to the range [0, 1]. For the BB task, the one-shot and two-shot performance of a 262M-parameter decoder-only transformer is assessed across five sub-tasks from (bench authors, 2023). The x-axis denotes the number of observed examples, while the y-axis reports task-specific metrics such as *multiple choice grad* or *exact string match*.

**Nano.** In addition to the scaling law benchmark, we incorporate data from nanoGPT-Bench and double descent (Nakkiran et al., 2021) to enhance the analysis of scaling behavior in the NLP domain, focusing on factors beyond the number of observed examples. nanoGPT-Bench (**Nano**) is a benchmark introduced in (Kadra et al., 2023), which evaluates the performance of nanoGPT trained on the OpenWebText dataset (Gokaslan et al., 2019) using 12 different hyperparameter configurations. Both the final and best performance are reported, resulting in a total of 24 scaling laws. In this benchmark, the x-axis corresponds to the model embedding size, while the y-axis represents either the final or best validation loss.

**ColPret.** Furthermore, we include the latest large dataset (**ColPret**; Choshen et al., 2024). The dataset aggregates scaling law data from many language models where the largest model is over 3 billion parameters and provides over 1000 estimated scaling laws. It consists of 485 LLMs, including GPT-3 (Brown, 2020), Mamba (Gu & Dao, 2023), Llama (Touvron et al., 2023b), etc., and covers more than 40 distinct model families. They report the number of epochs and computational cost (FLOPs) additionally, but we focus on the number of seen tokens, leading to 591 scaling laws. We filter scaling laws based on their length, keeping only those between 10 and 1000, resulting in 194 scaling laws. Additionally, we further exclude two scaling laws: *pythia-19m-deduped* due to excessive noise and *rpj-d=1024_l=24_h=8-0.25* because its loss saturates. Since we remove two more scaling laws, the final dataset consists of 192 scaling laws. The x-axis in this dataset represents the number of seen tokens, while the y-axis corresponds to either the loss or perplexity.

**DD.** The study by Nakkiran et al. (2021) introduces scaling laws that exhibit a *double descent* pattern, challenging the conventional assumption of unidirectional performance trends. The researchers trained and evaluated CNNs, ResNets, and Transformers on datasets such as CIFAR-10, CIFAR-100 (Krizhevsky et al., 2009), IWSLT'14 de-en (Cettolo et al., 2012), and WMT'14 en-fr (Bojar et al., 2014), exploring various combinations of optimizer settings, augmentations, and noise levels. In this work, we incorporate 16 publicly available scaling laws from (Nakkiran et al., 2021) that illustrate double descent behavior (**DD**). In these scaling laws, the x-axis represents either the model embedding size or the number of observed examples, while the y-axis reflects the test error or cross-entropy test loss.

## C. Implementation Details

$\mathcal{M}_{1-4}$. For experiments involving $\mathcal{M}_{1-4}$, we refer to the codebase provided by Alabdulmohsin et al. (2022)[2]. We fit $\mathcal{M}_{1-4}$ directly to the original scaling laws without normalization.

---

[2]The implementation for $\mathcal{M}1-4$ is available at: https://github.com/google-research/google-research/tree/master/revisiting_neural_scaling_laws.

**Broken Neural Scaling Laws (BNSL).** We obtained the code from the BNSL (Caballero et al., 2022) authors. The implementation of the code has minor adjustments to ensure stable and efficient training, as detailed below:

$$\text{BNSL: } y = \exp\left( \log\left(e^{f_{\text{bnsl}}(\tilde{x})} + e^a\right) + y_{\text{mean}} \right),$$

$$\text{where } f_{\text{bnsl}}(\tilde{x}) = W_3 \cdot \left( W_2 \cdot \text{softplus}(W_1 \tilde{x} + b_1) + b_2 \right) + b_3,$$

$$\tilde{x} = \frac{\log(x) - \mu_x}{\sigma_x},$$

$$\mu_x = \text{mean}(\log(x_{\text{train}})), \quad \sigma_x = \text{std}(\log(x_{\text{train}})). \quad (11)$$

The provided code lacked information on determining the number of breaks, deciding whether to crop or selecting cropping points. Despite our follow-up requests for clarification, we did not receive a response. Consequently, we designed and tested various scenarios based on the descriptions in the paper.

To determine the optimal number of breaks, we validate using either the last 10% or the final point of the context, selecting the number of breaks that produced the best performance. When the optimal number of breaks is greater than zero, we further explore cropping. Specifically, we identify five evenly spaced points within the context as potential crop locations and apply the crop that yielded the best validation performance. We then apply the crop that yields the best validation performance to our inference process as well.

Table 10: **BNSL RMSLE across implementations**. † indicates that some trials failed, and these values were excluded.

| Validation split | Crop | IC | NLP, Nano | DD |
|---|---|---|---|---|
| Last 10% | ✗ | 0.0406 | 0.0223 | 0.0468 |
| Final point | ✗ | 0.0432 | 0.0186† | 0.0379 |
| Last 10% | ✓ | 0.0576 | 0.0231 | 0.0477 |
| Final point | ✓ | 0.0571 | 0.0228† | 0.0987 |

We summarize the results of four experiments in Table 10, each with variations in the validation set configuration and cropping application. Similar to $\mathcal{M}_{1-4}$, no normalization is applied during the training of BNSL. Due to the large size of ColPret, we focus on IC, NLP, and DD. While the combination of validation using the final point and no cropping yielded the best performance in the NLP and DD tasks, some predictions in the NLP task failed. Consequently, in the main text, we report results based on validation using the last 10% of the context without cropping.

Table 11: **MCMC prior distribution**.

| Name | Function | Prior |
|---|---|---|
| $\mathcal{M}_1$ | $y = ax^{-b}$ | $a \sim \mathcal{U}(-1, 0.5) \quad \log b \sim \mathcal{N}(-2, 1)$ |
| $\mathcal{M}_2$ | $y = ax^{-b} + c$ | $a \sim \mathcal{U}(0, 1.5) \quad \log b \sim \mathcal{N}(-2, 1) \quad c \sim \mathcal{U}(0, 1)$ |
| $\mathcal{M}_3$ | $y = a(x^{-1} + d)^b$ | $a \sim \mathcal{U}(0, 1) \quad \log b \sim \mathcal{N}(-2, 1) \quad \log d \sim \mathcal{U}(0, 1)$ |
| $\mathcal{M}_4$ | $x = g(y) = \left(\frac{y}{a(1-y)^{\alpha}}\right)^{-\frac{1}{b}}$ | $\log a \sim \mathcal{U}(1, 1000) \quad \log b \sim \mathcal{N}(-1, 1) \quad \log \alpha \sim \mathcal{N}(0, 1)$ |
| BNSL | $y = \exp\left(\log\left(e^{f_{\text{bnsl}}(\tilde{x})} + e^a\right) + y_{\text{mean}}\right),$ where $f_{\text{bnsl}}(\tilde{x}) = W_3 \cdot (W_2 \cdot \text{softplus}(W_1 \tilde{x} + b_1) + b_2) + b_3$ | $a \sim \mathcal{N}(0, 5) \quad W \sim \mathcal{N}(0, 5) \quad b \sim \mathcal{N}(0, 5)$ |

**Markov Chain Monte Carlo (MCMC).** We implement MCMC using EMCEE (Foreman-Mackey et al., 2013). The parameters for EMCEE are configured as follows: the number of walkers in the ensemble (`nwalkers`) is set to 30, the number of samples (`nsamples`) is set to 150 (except for Fig. 6), the part of the chain omitted to account for mixing (`burn_in`) is set to 50, and the sub-sampling frequency (`thin`) is set to 1. Appropriate prior ranges are defined for each functional form, as detailed in Table 11. The number of breaks for MCMC (BNSL) is determined through validation, following the same approach as used for point estimation.

During MCMC training, the x-values are normalized by dividing them by their maximum value, scaling the maximum to 1. However, for MCMC ($\mathcal{M}_4$), where smaller x-values frequently lead to failures, the x-values are instead normalized to a maximum of 1000. For the DD and Nano datasets, the y-values are normalized by dividing them by the maximum context value.

Table 12: **BLR performance with non-NN basis functions**

| Basis | IC AVG | | NLP AVG | | ColPret | | DD | |
|---|---|---|---|---|---|---|---|---|
| | RMSLE ($\downarrow$) | LL ($\uparrow$) | RMSLE ($\downarrow$) | LL ($\uparrow$) | RMSLE ($\downarrow$) | LL ($\uparrow$) | RMSLE ($\downarrow$) | LL ($\uparrow$) |
| Polynomial | 3.979 | -6.71 | 1.311 | -57.65 | 2.573 | -174.43 | 2.420 | -408.05 |
| RBF | 2.118 | -34.82 | 0.253 | -38.54 | 0.633 | -34.82 | 1.085 | -11.68 |
| Fourier | 1.362 | -56.52 | 0.283 | -160.41 | 0.529 | -52.46 | 1.801 | -34.99 |
| Sigmoid | 8.819 | -3.37 | 0.187 | -70.81 | 1.324 | -4.54 | 1.419 | -6.51 |
| Spline | 0.551 | -48.56 | 0.159 | -53.09 | 0.241 | -64.49 | 0.238 | -69.89 |

**Bayesian Linear Regression (BLR) and Deep Kernel Gaussian Process (DKGP).** We conducted additional experiments with the following baselines: BLR (Bishop, 2006) with neural network basis functions, BLR with polynomial basis functions, BLR with RBF basis functions, BLR with Fourier basis functions, BLR with sigmoid basis functions, BLR with spline basis functions, and DKGP (Wilson et al., 2016). Hyperparameters, including those of the neural network used as the basis function for BLR, are tuned via marginal log-likelihood. We implement BLR using `BayesianRidge` and DKGP using `GPyTorch`. The feature dimension is fixed to 5 for all additional baselines, and all target values $y$ are scaled using `RobustScaler`, fitted on the context points ($\forall y \in \mathcal{C}$). The neural networks used for BLR with neural network basis functions and DKGP are trained for 1000 full-batch epochs using the Adam optimizer (Kingma, 2014) with a learning rate of 0.01. As shown in Table 12, BLR models with predefined basis functions exhibit significant performance degradation.

**LC-PFN** For our experiments, we utilize the LC-PFN as described in Adriaensen et al. (2023)[3]. Since the range of x for the prior was sampled from [0, 100] during LC-PFN training, we normalize x-values to the same range during inference. LC-PFN includes its own normalization method for y-values, enabling it to predict learning curves across various ranges and directions. However, when evaluating LC-PFN on scaling laws, we found that its normalization method failed for certain scaling laws in the LM task. To address this, we apply the NSL-PFN normalization method, followed by a horizontal flip, *i.e.*, $y \leftarrow 1 - y$. This adjustment reduced the RMSLE for the LM task from 0.5254 to 0.1151. We report LC-PFN performance using this updated methodology.

**NSL-PFN** We employ a Transformer (Vaswani, 2017), representing each context pair $(x, y)$ and the target $x$ as individual tokens while excluding positional encoding. These tokens are encoded with a simple linear layer. The target tokens are assumed to be conditionally independent of each other, *i.e.*, we apply masking to the attention matrix so that each token only attends to the context tokens.

Following Müller et al. (2021), we discretize the output distribution into a finite number of bins. The size (width) of each bin is set so that, under the prior, has an equal probability of falling in each bin. The number of bins is set to 1,000. For RMSLE, we use the median of output distribution.

NSL-PFN inherits the hyperparameters of architecture from PFNs, such as the number of layers (`nlayers`=12), number of heads (`nheads`=4), and the size of each hidden layer (`nhidden`=512). We train NSL-PFN on 1.6M scaling laws sampled from the prior for 100K iterations, *i.e.*, mini-batch size is set to 16, with Adam optimizer (Kingma, 2014). The learning rate is set to 0.00002 with cosine annealing, and the warm-up phase spans the first 25K iterations. To determine the size of $\mathcal{D}$ (*i.e.*, $M + N$), we uniformly sample from the log space within the range $[50, 500]$. The training time is roughly 2.6 hours on NVIDIA A100-SXM4-80GB.

Since NSL-PFN is trained with priors input ranges of [0.01, 1] for x and [0, 1] for y, we preprocess the evaluation data accordingly. Specifically, we normalize x-values to [0.01, 1]. For y-values, no additional preprocessing is required for the scaling laws benchmark datasets (IC, NMT, LM, and BB), as they are already scaled to [0, 1]. However, for the Nano and DD datasets, where y-values exceed 1, we normalize all y-values by dividing them by the largest value in their respective contexts. This ensures that the context part is normalized to the [0, 1] range, although values in the target part may still exceed 1.

---

[3]The LC-PFN implementation is available at: https://github.com/automl/lcpfn.

## D. Hyperparameter Tuning for Prior Distributions

Table 13: Hyperparameter search space for Bayesian optimization. For each "Log-Parameter" of a given "Model" (*e.g.*, $\log a$ for $\mathcal{M}_3$), its normal prior distribution $\mathcal{N}(\mu, \sigma^2)$ was refined. This was done by optimizing the "Prior Statistic"—specifically the prior's mean ($\mu$) or standard deviation ($\sigma$). The "Search Range" column defines the optimization bounds for these statistics. Default prior values are in Table 1.

| Model | Log-Parameter | Prior Statistic | Search Range |
|---|---|---|---|
| $\mathcal{M}_3$ | $\log a$ | Mean ($\mu$) | Real(-2.0, 0.0) |
| | | Std. Dev. ($\sigma$) | Real(0.1, 1.5) |
| | $\log(-b)$ | Mean ($\mu$) | Real(-3.0, -1.0) |
| | | Std. Dev. ($\sigma$) | Real(0.1, 1.5) |
| | $\log d$ | Mean ($\mu$) | Real(-0.5, 0.5) |
| | | Std. Dev. ($\sigma$) | Real(0.1, 1.5) |
| $\mathcal{M}_4$ | $\log a$ | Mean ($\mu$) | Real(-0.5, 0.5) |
| | | Std. Dev. ($\sigma$) | Real(0.1, 1.5) |
| | $\log(-b)$ | Mean ($\mu$) | Real(-2.0, 0.0) |
| | | Std. Dev. ($\sigma$) | Real(0.1, 1.5) |
| | $\log \alpha$ | Mean ($\mu$) | Real(-0.5, 0.5) |
| | | Std. Dev. ($\sigma$) | Real(0.1, 1.5) |

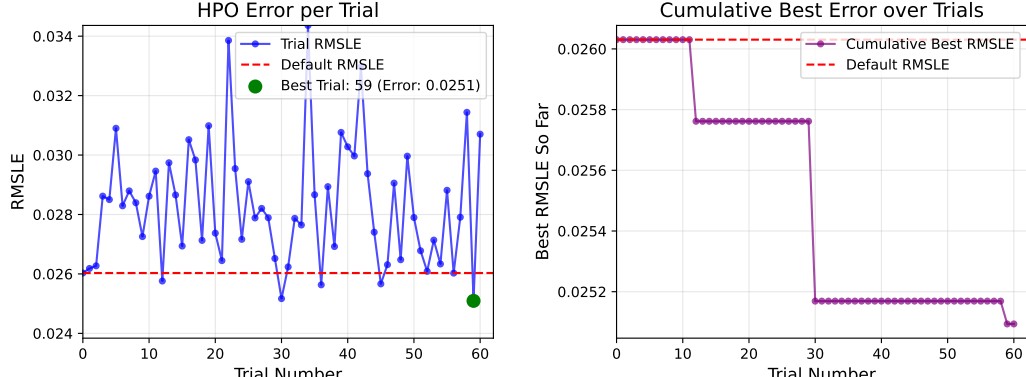

Figure 7: Results of Bayesian Optimization for prior distribution hyperparameters over 60 trials. Top: RMSLE per trial, with the default RMSLE shown as a dashed red line and the best trial highlighted. Bottom: Cumulative best RMSLE found over trials, compared against the default RMSLE.

Our proposed methodology utilizes several functional forms (e.g., $\mathcal{M}_3$, $\mathcal{M}_4$) for modeling scaling law segments. The prior distributions for the parameters of these functions are detailed in the main text in Table 1. We conduct a simple Bayesian optimization (BO) on the prior parameters to minimize the average RMSLE over 60 BO steps. The HPO targeted the location (mean) and scale (standard deviation) parameters of the Normal prior distributions for the logarithmic parameters of $\mathcal{M}_3$ and $\mathcal{M}_4$. The specific search ranges for these hyperparameters are detailed in Table 13. The results of the Bayesian Optimization are illustrated in Fig. 7. The BO process successfully identified prior parameter configurations that yielded slight improvements in performance over our default settings. This outcome suggests that our initial procedure for selecting prior parameters was not overly aggressive and that there remains some scope for further marginal enhancements through systematic optimization of these prior hyperparameters.

## E. Visualization

From the next page, we present additional extrapolation visualizations.

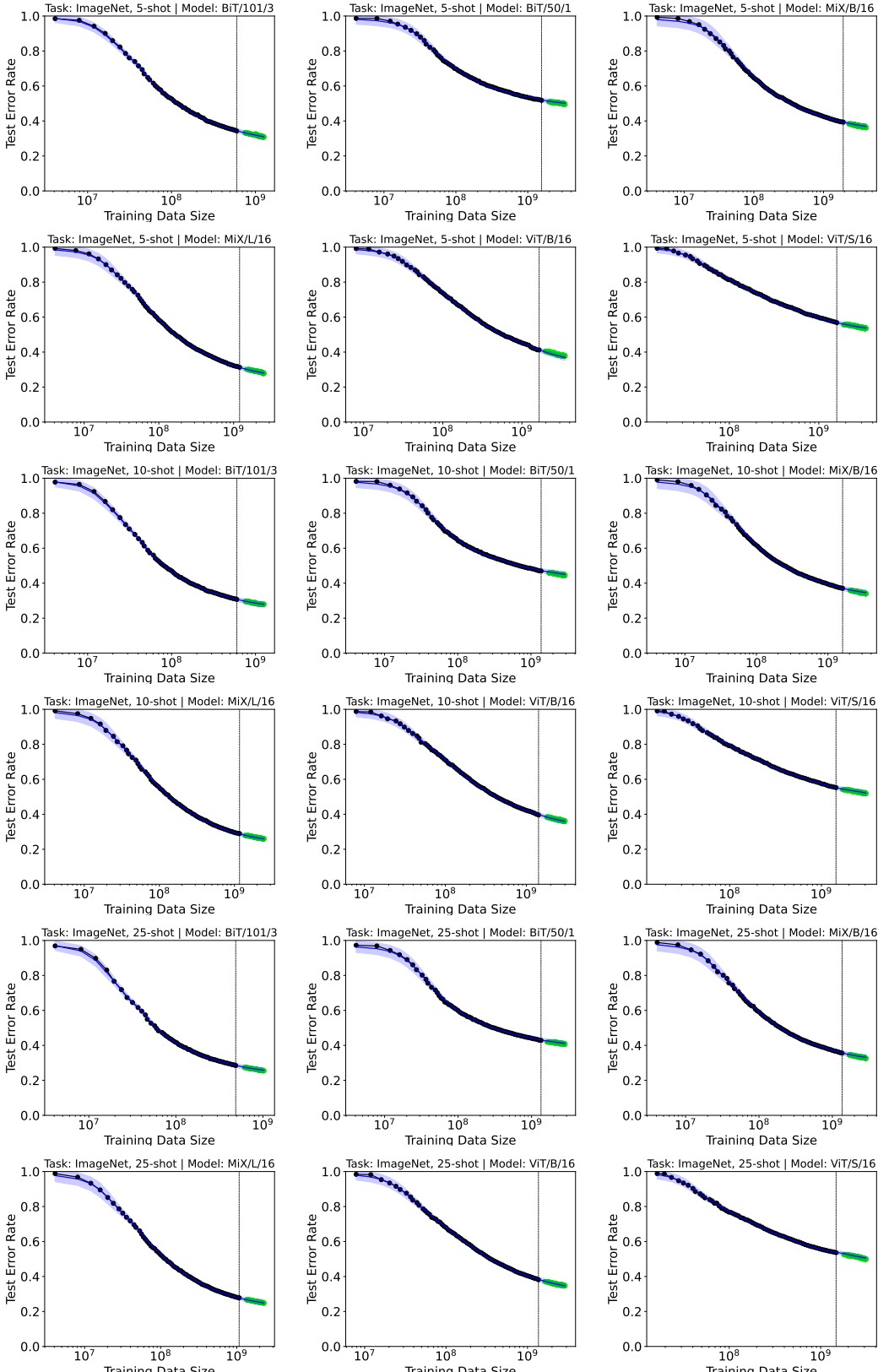

Figure 8: **Visualization of extrapolation results on ImageNet**.

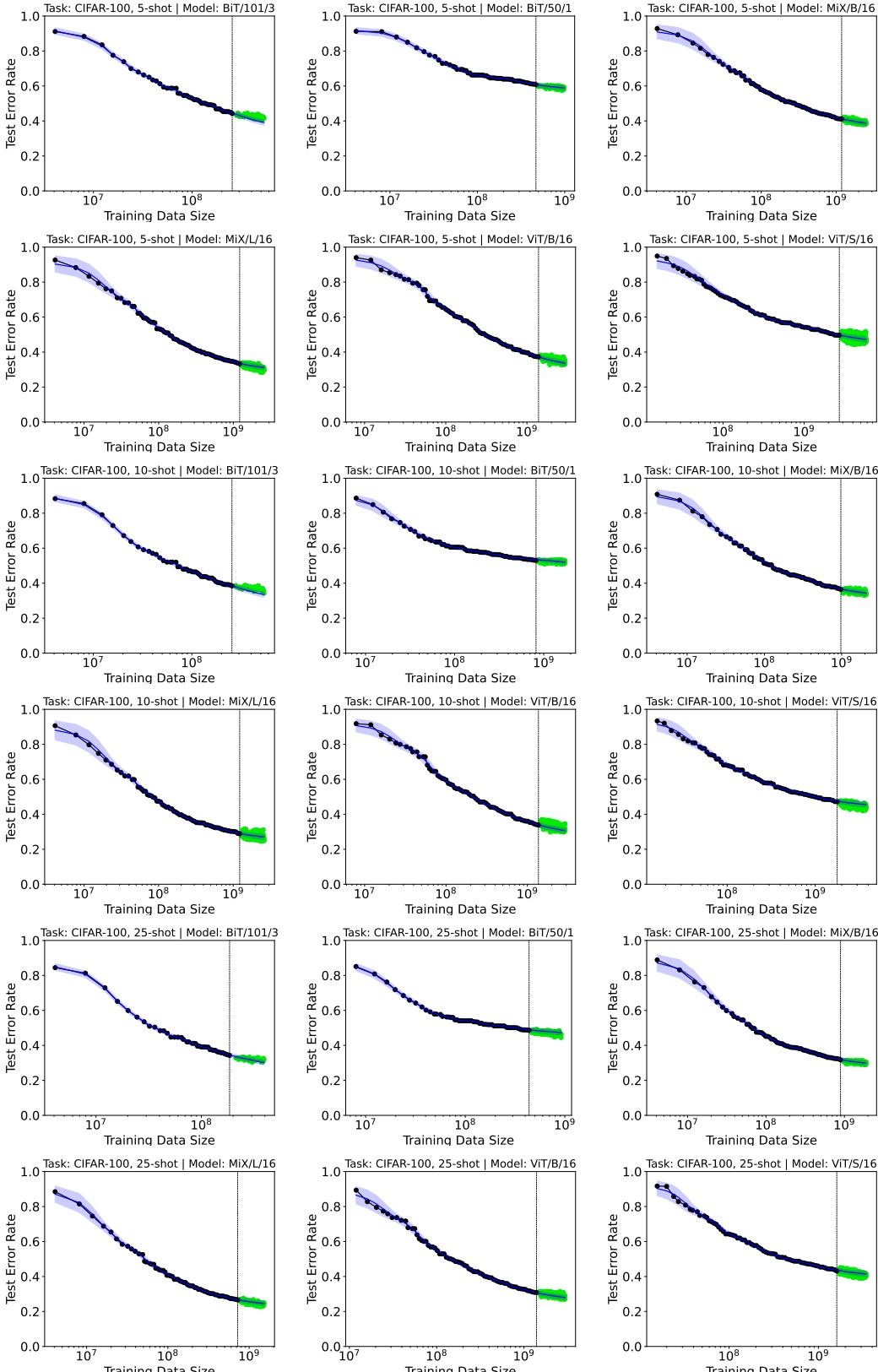

Figure 9: **Visualization of extrapolation results on CIFAR-100**.

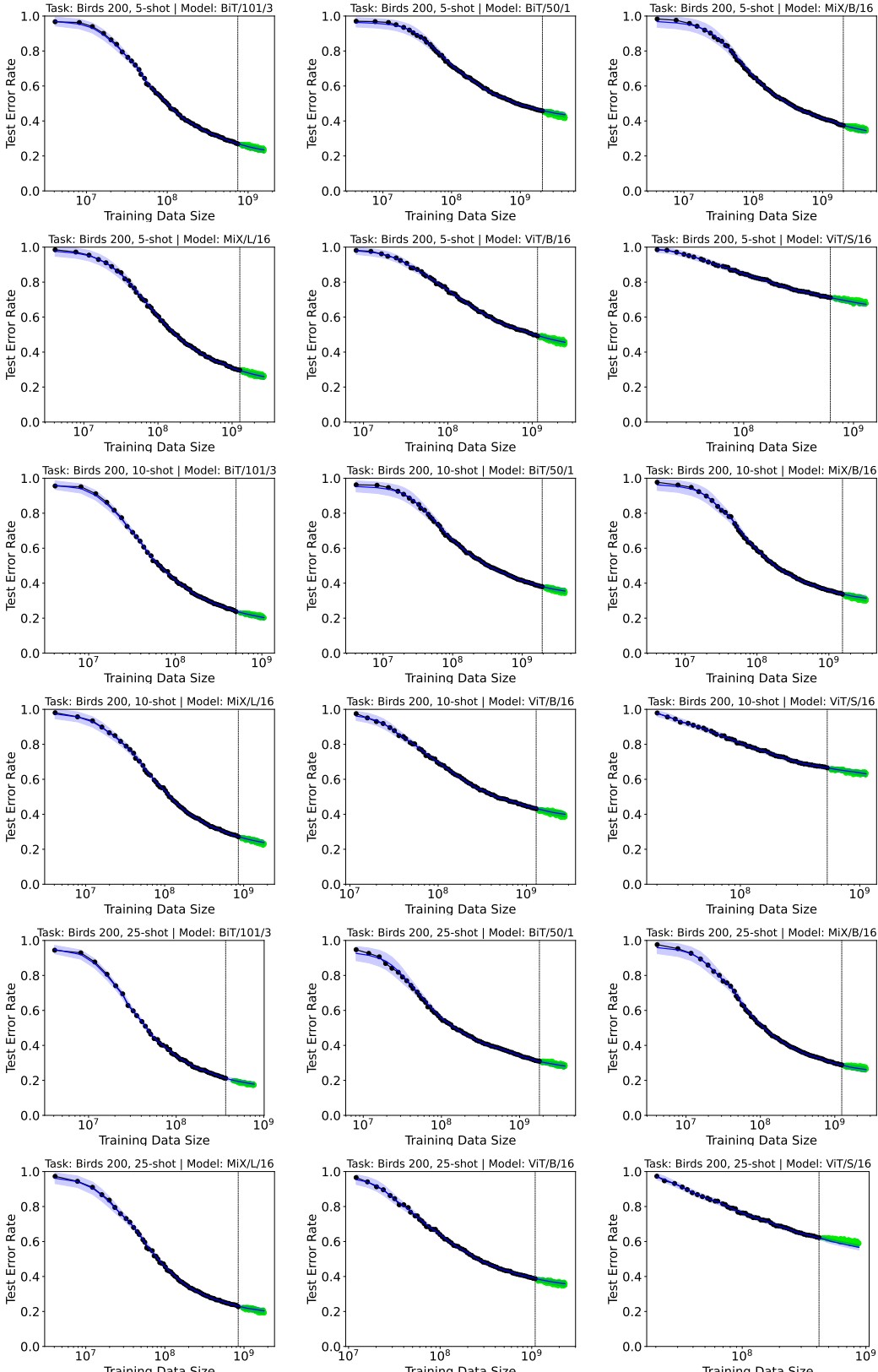

Figure 10: **Visualization of extrapolation results on Birds 200**.

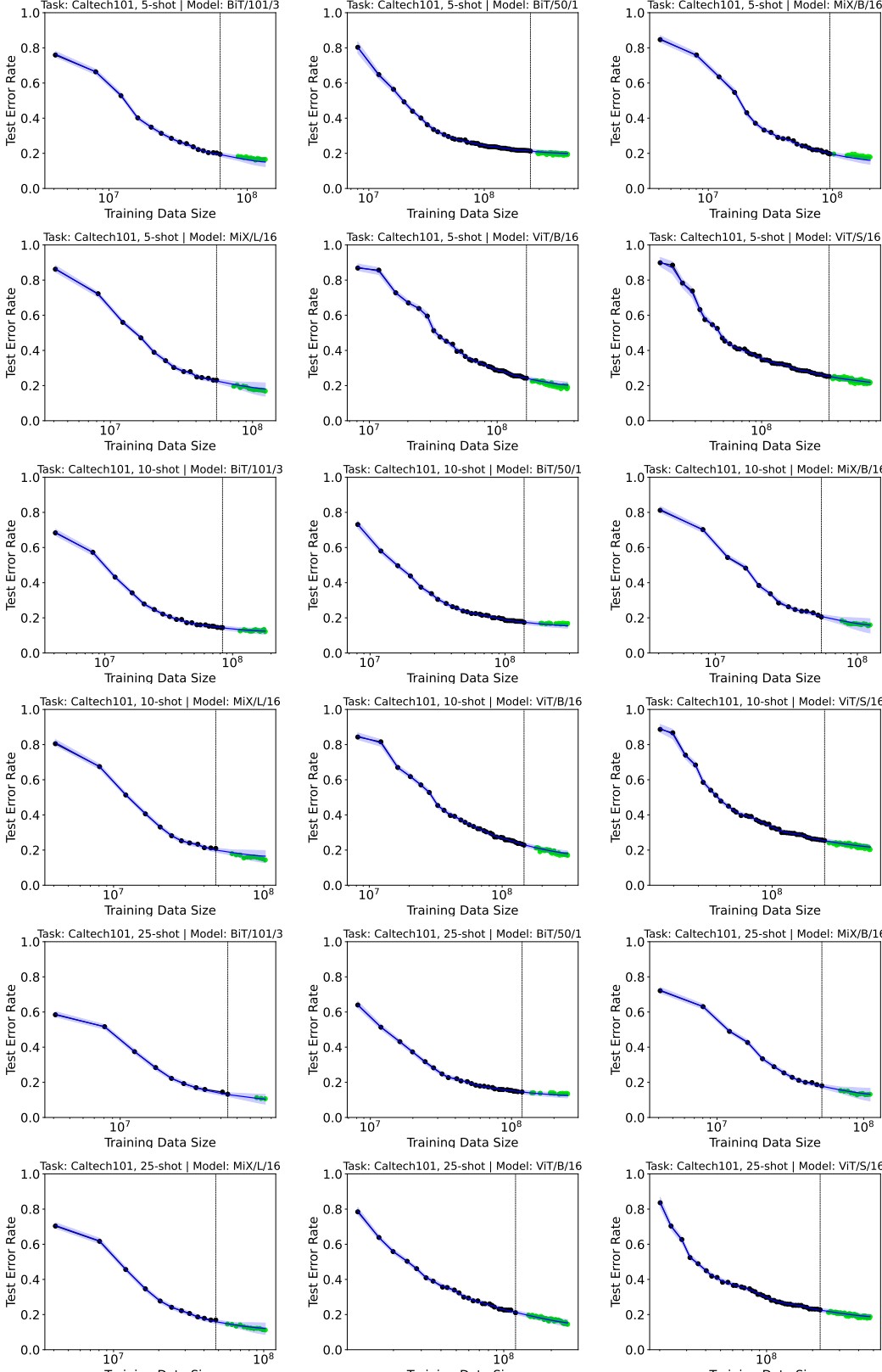

Figure 11: **Visualization of extrapolation results on Caltech101**.

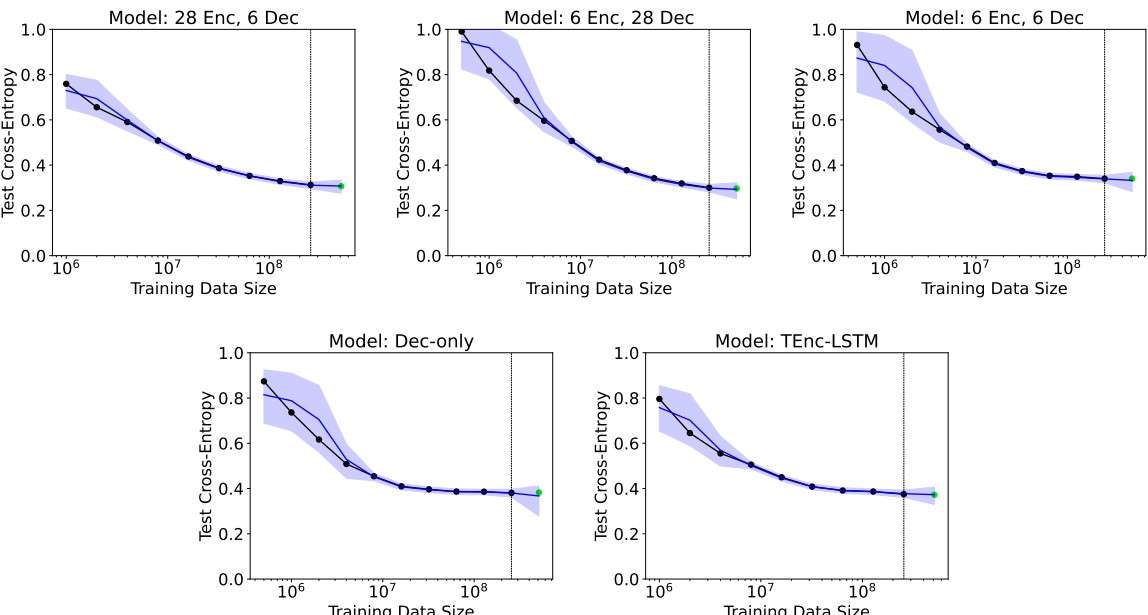

Figure 12: **Visualization of extrapolation results on neural machine translation (NMT).**

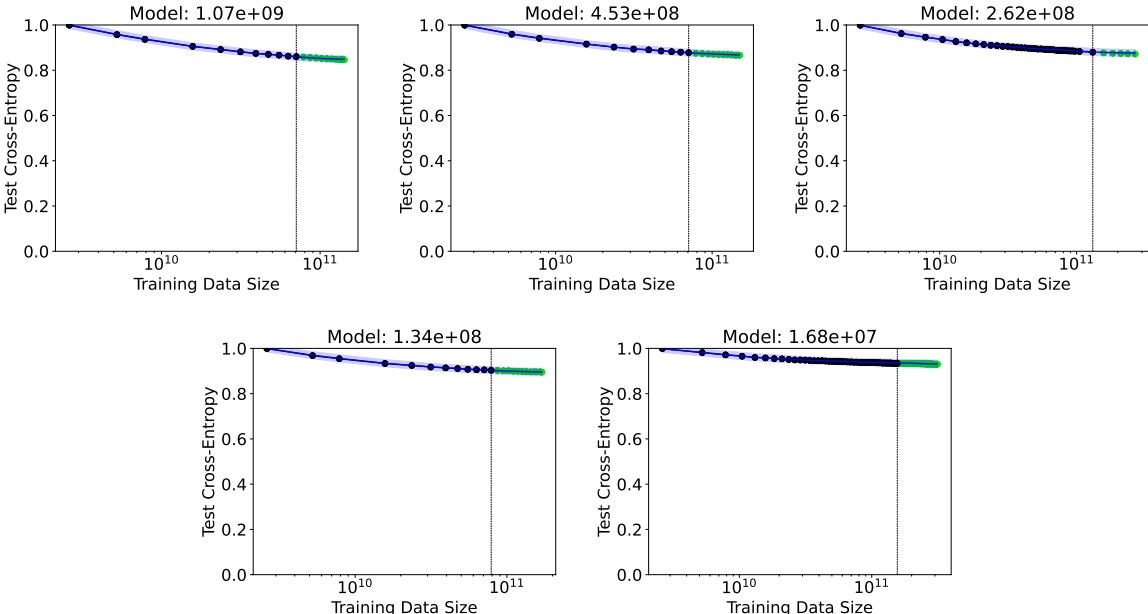

Figure 13: **Visualization of extrapolation results on language modeling (LM).**

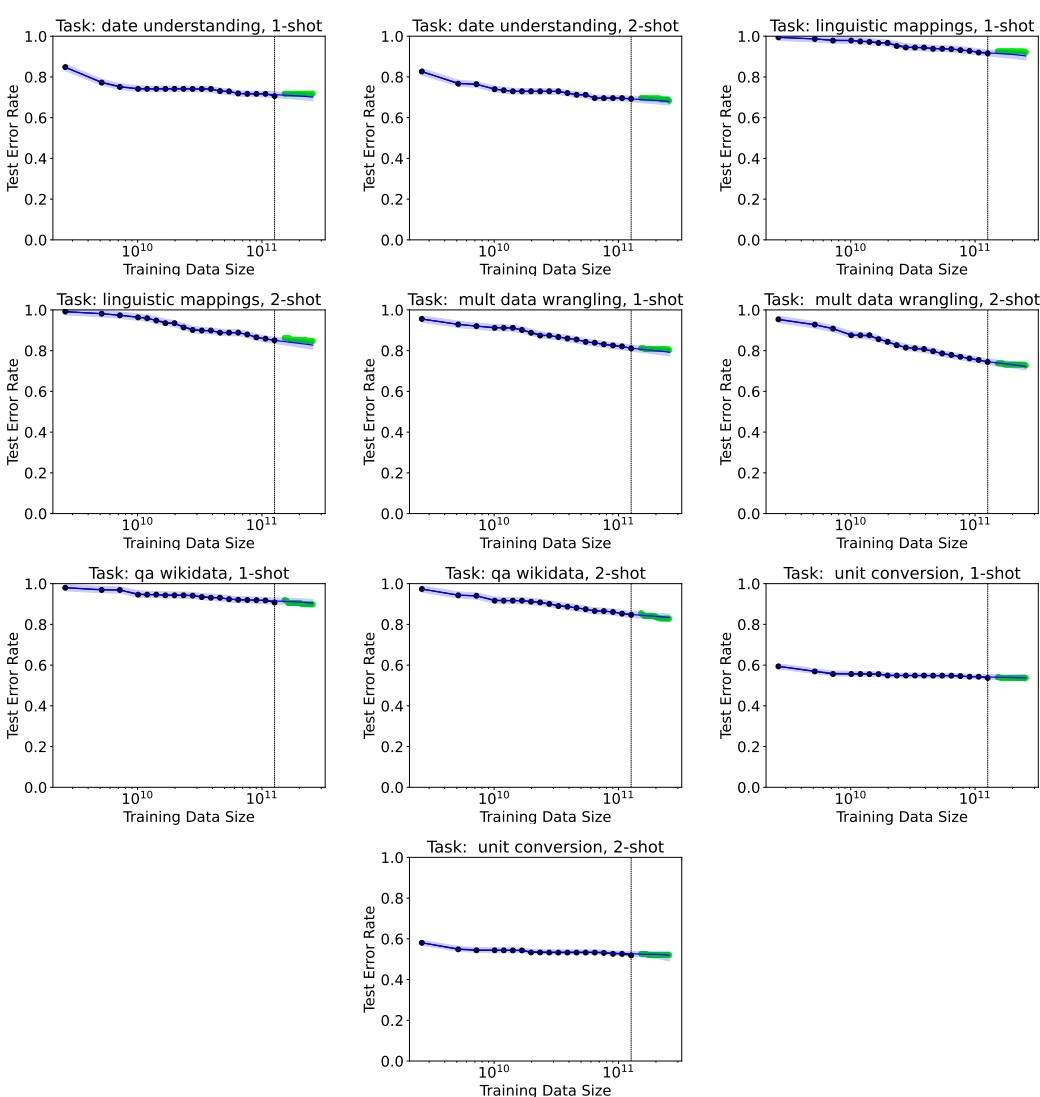

Figure 14: **Visualization of extrapolation results on Big-Bench (BB)**.

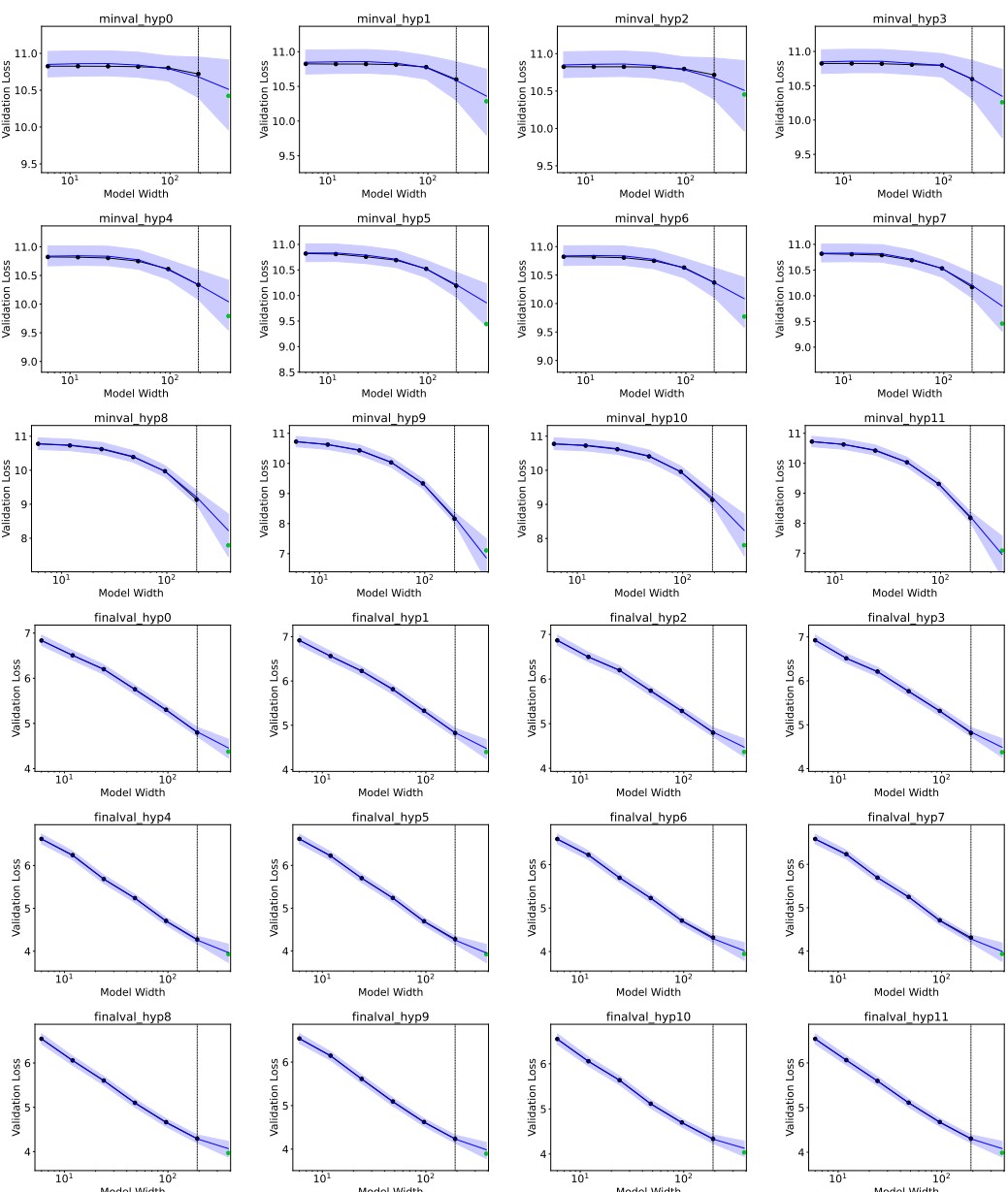

Figure 15: **Visualization of extrapolation results on NanoGPT-Bench (Nano)**.

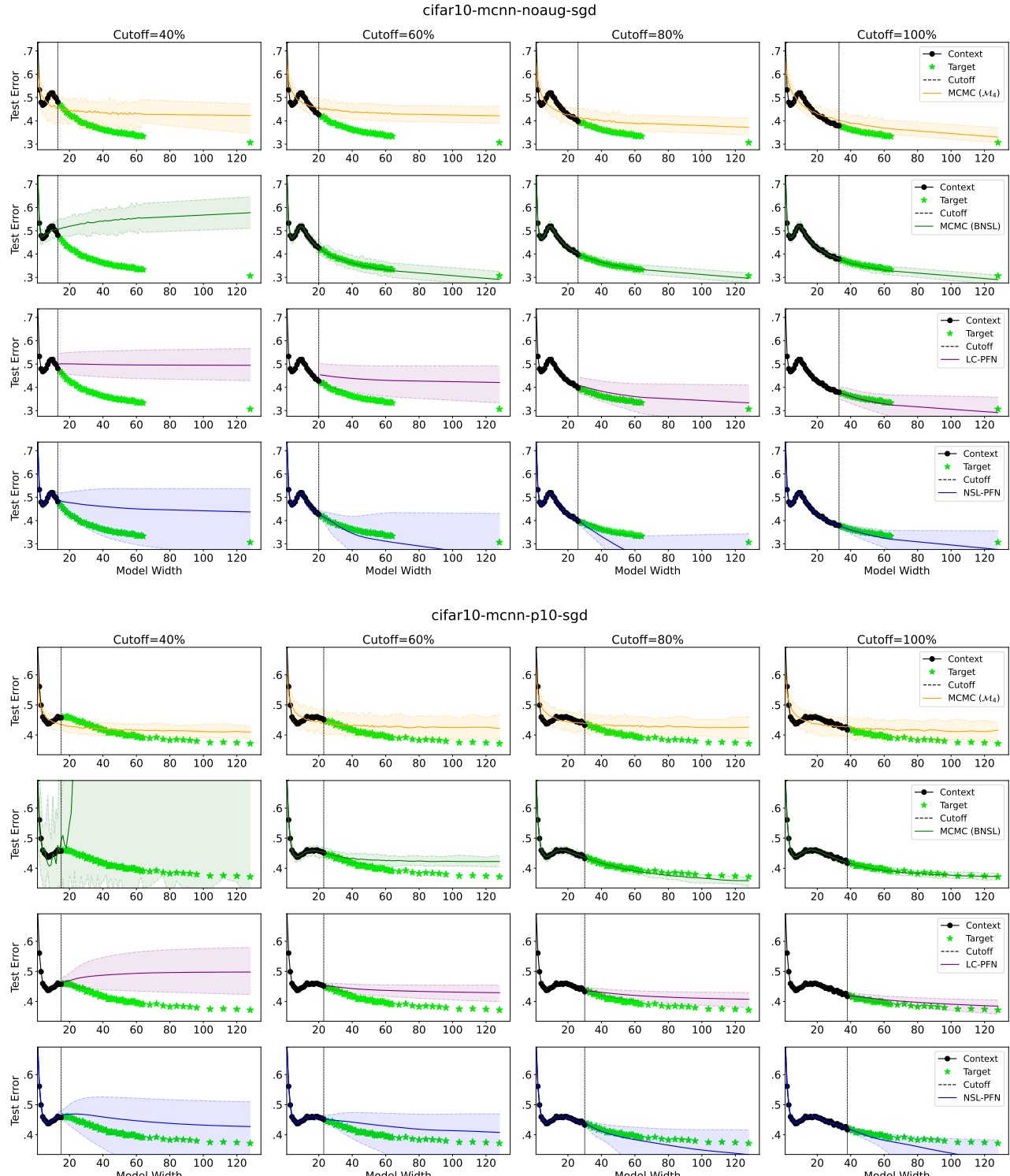

Figure 16: **Visualization of extrapolation results on double descent (DD; Nakkiran et al., 2021), Part 1 of 8.**

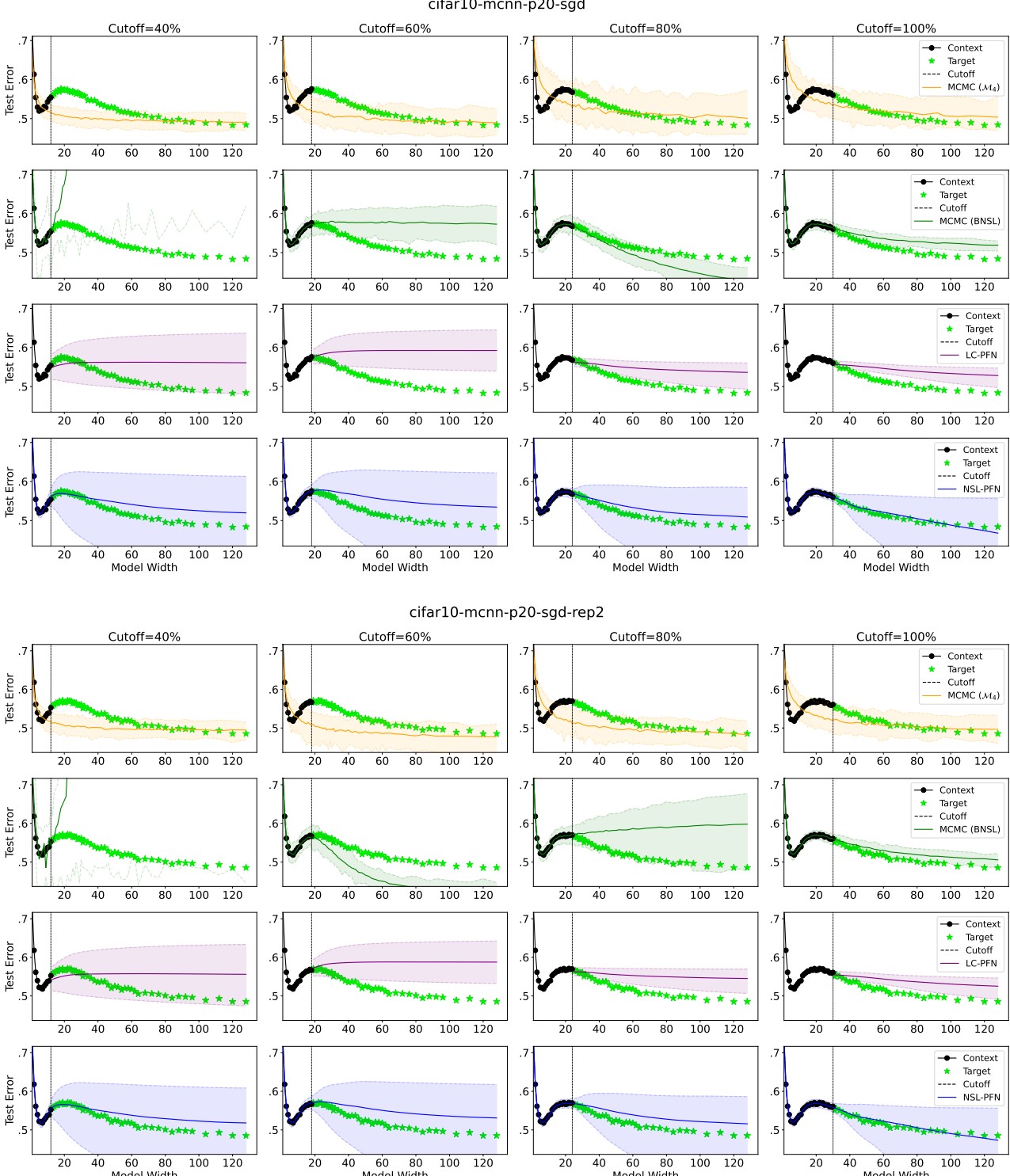

Figure 17: **Visualization of extrapolation results on double descent (DD; Nakkiran et al., 2021), Part 2 of 8.**

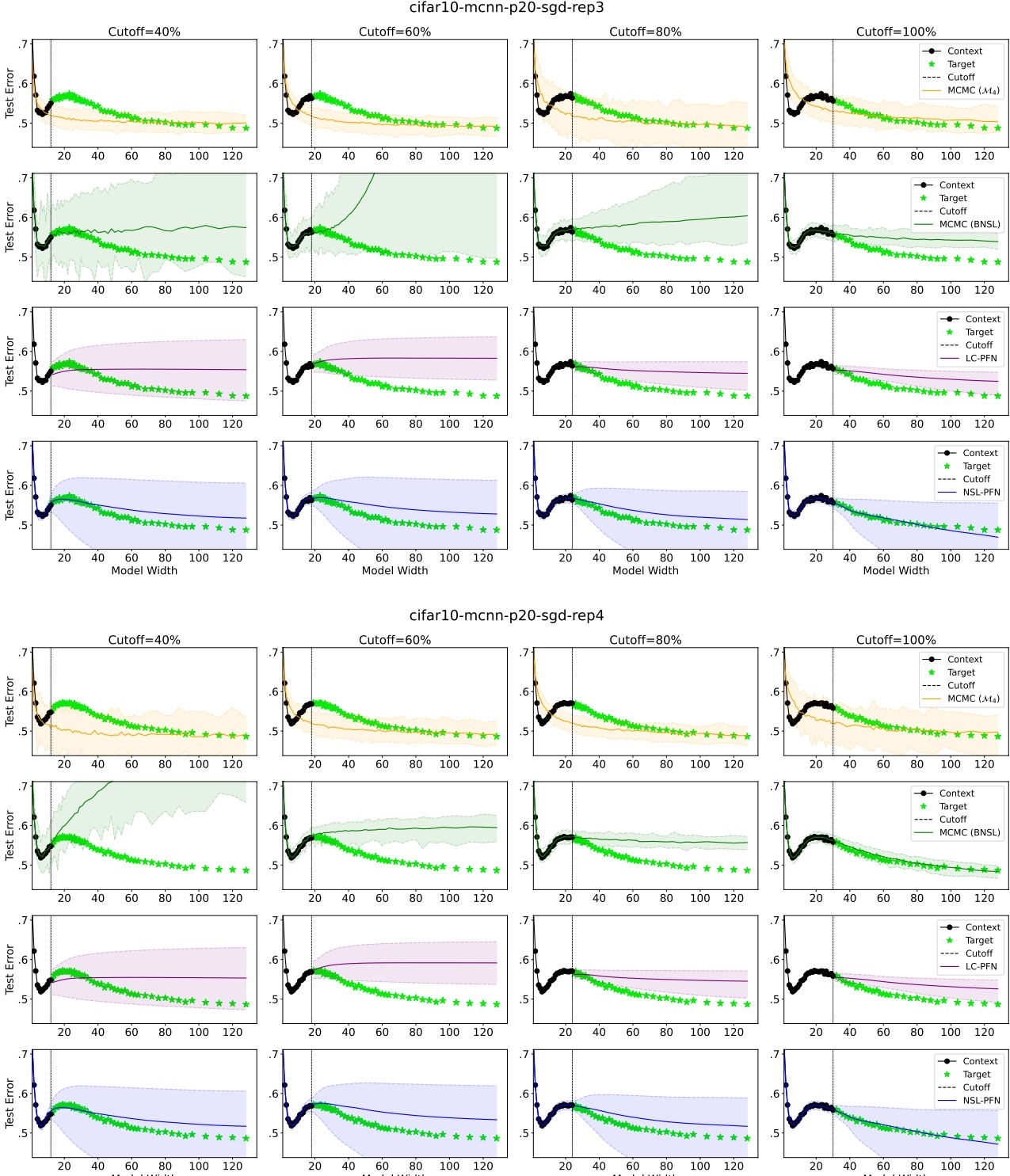

Figure 18: **Visualization of extrapolation results on double descent (DD; Nakkiran et al., 2021), Part 3 of 8.**

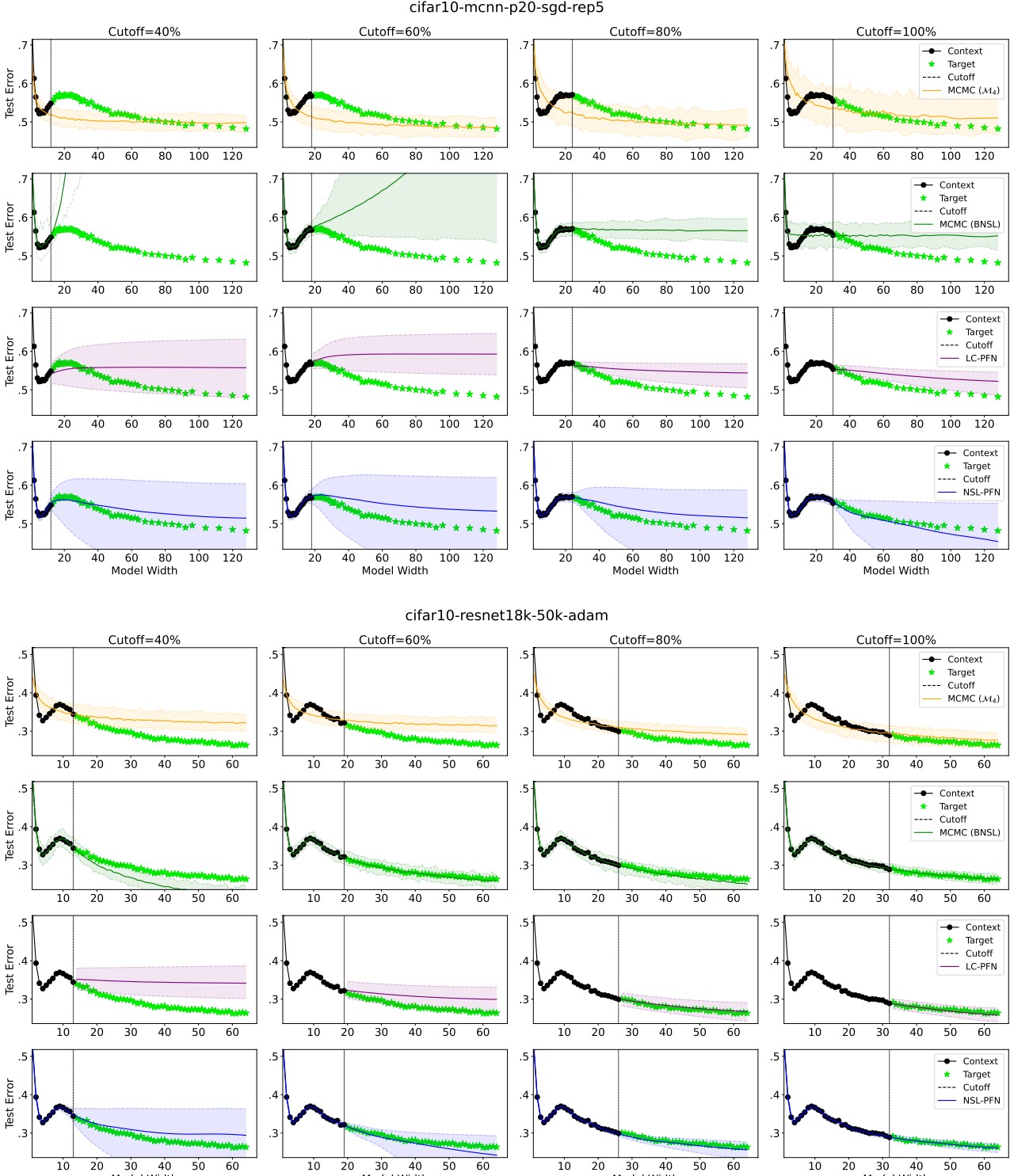

Figure 19: **Visualization of extrapolation results on double descent (DD; [Nakkiran et al., 2021](#)), Part 4 of 8.**

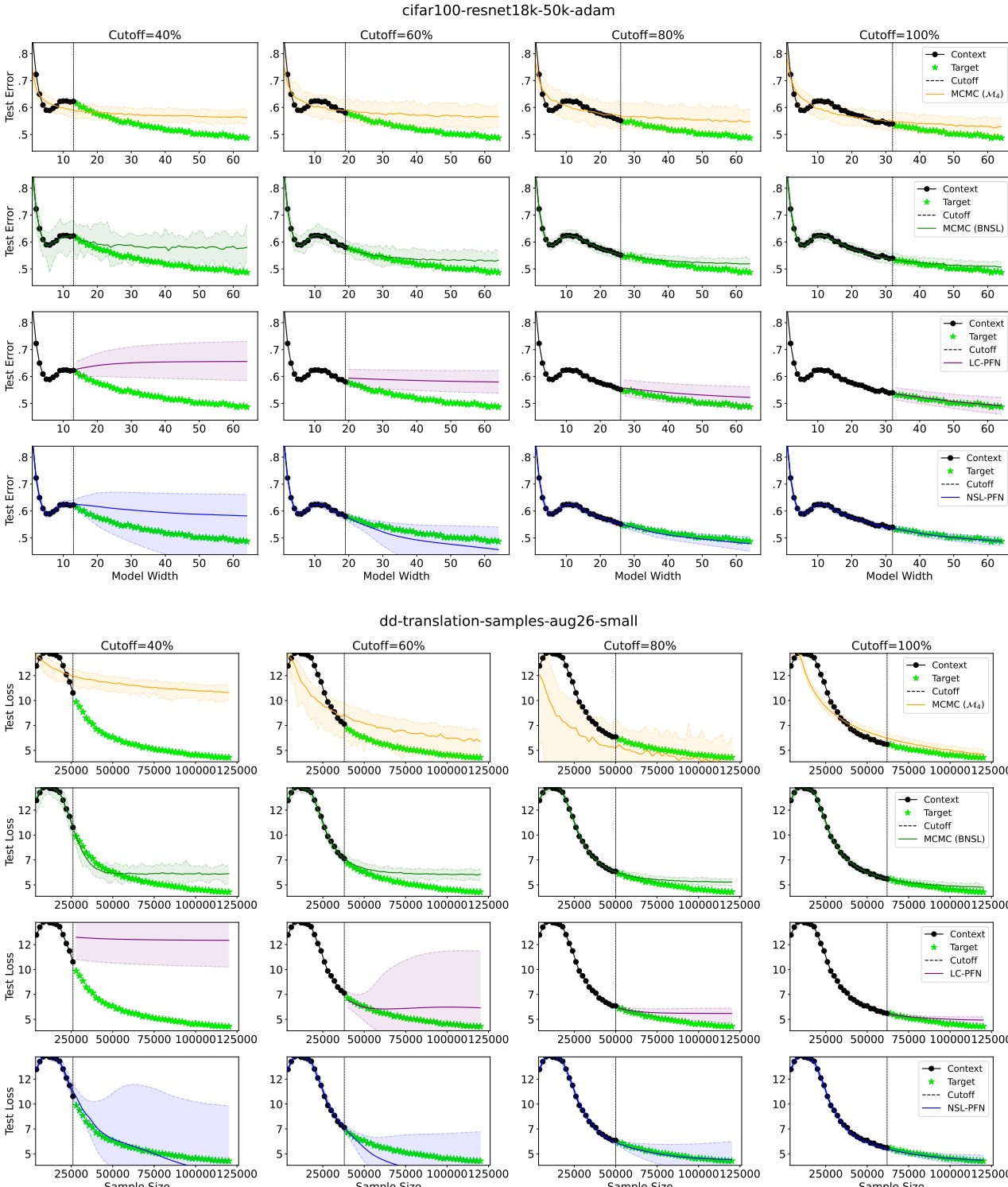

Figure 20: **Visualization of extrapolation results on double descent (DD; Nakkiran et al., 2021), Part 5 of 8.**

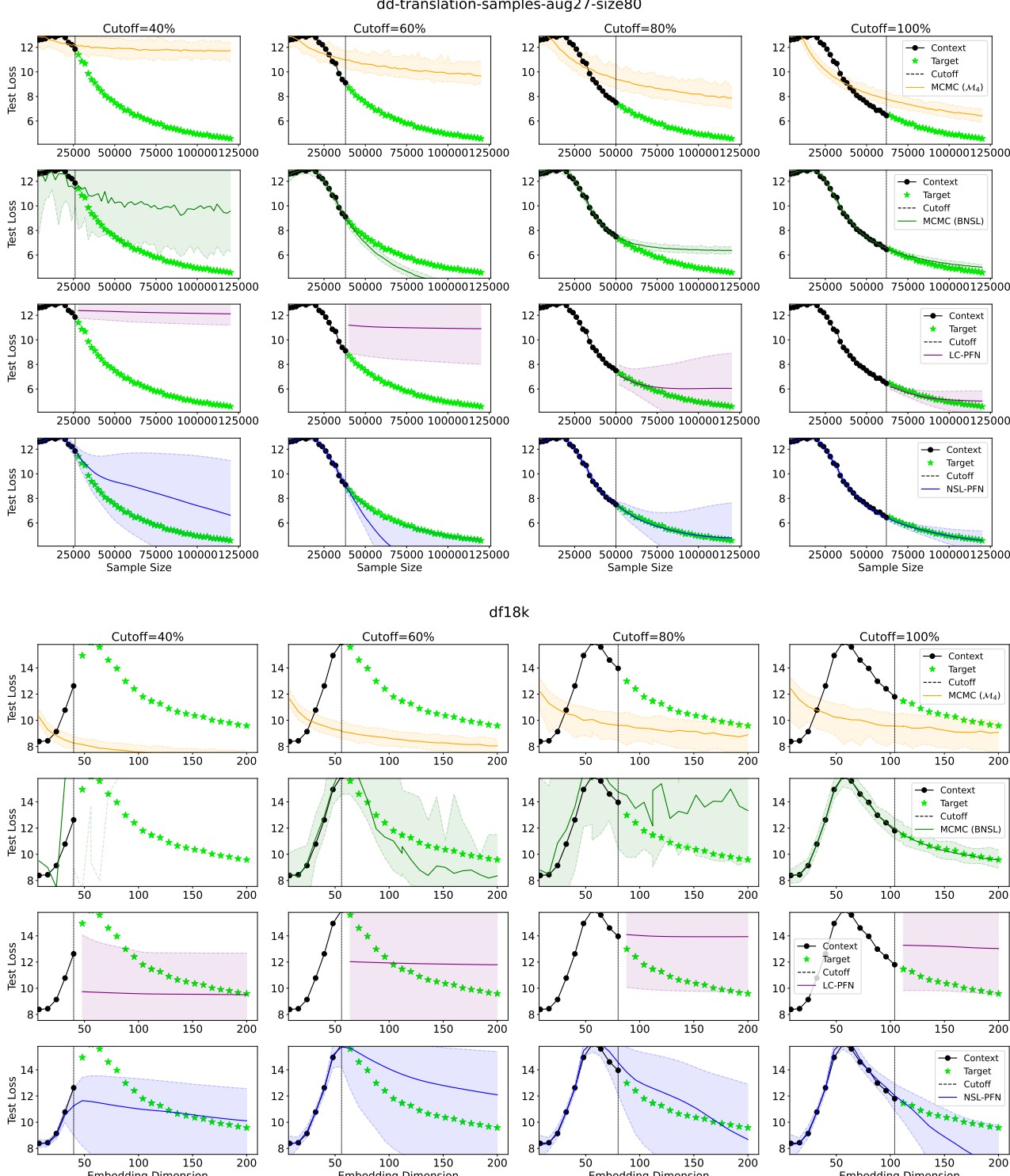

Figure 21: **Visualization of extrapolation results on double descent (DD; Nakkiran et al., 2021), Part 6 of 8.**

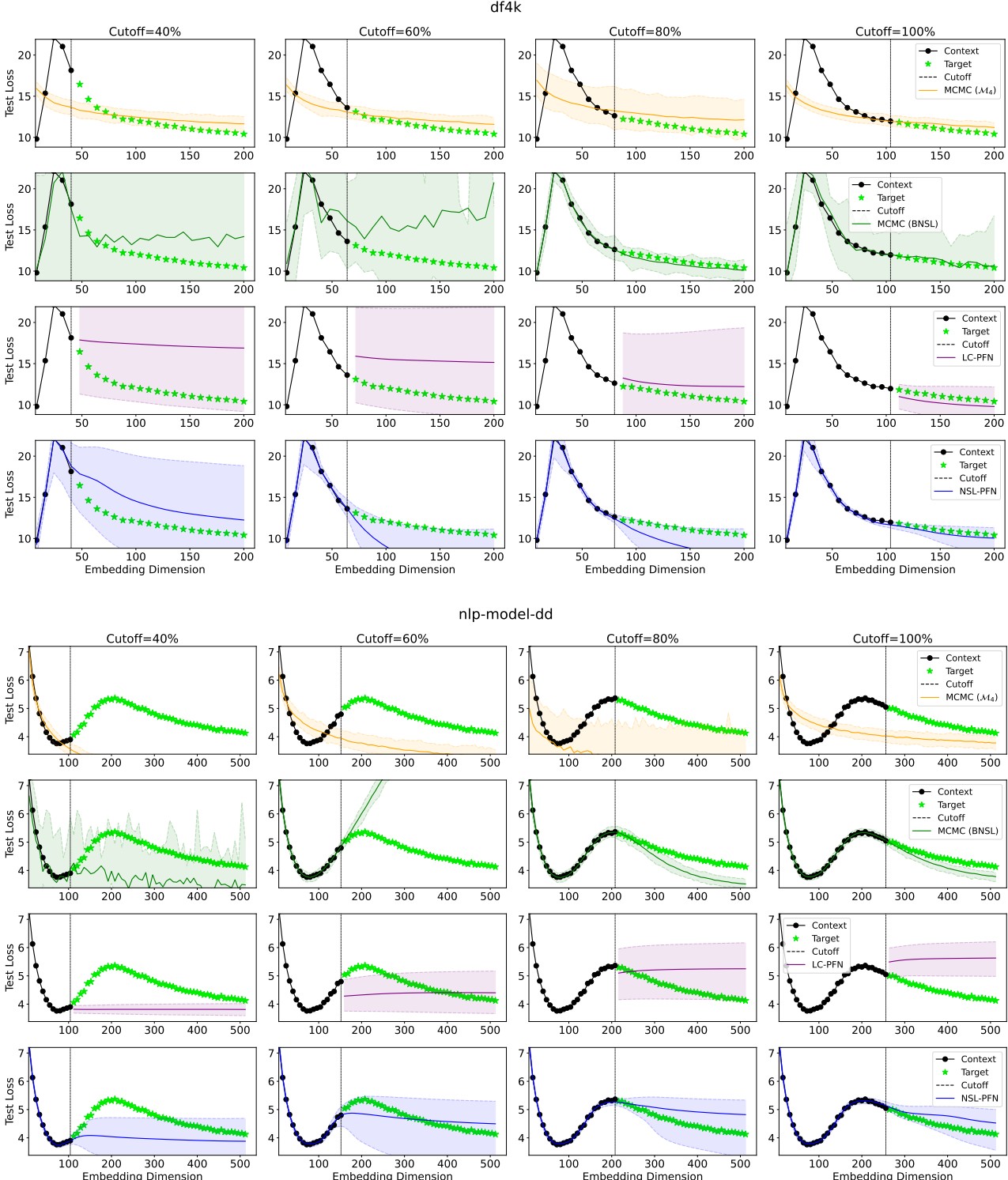

Figure 22: **Visualization of extrapolation results on double descent (DD; Nakkiran et al., 2021), Part 7 of 8.**

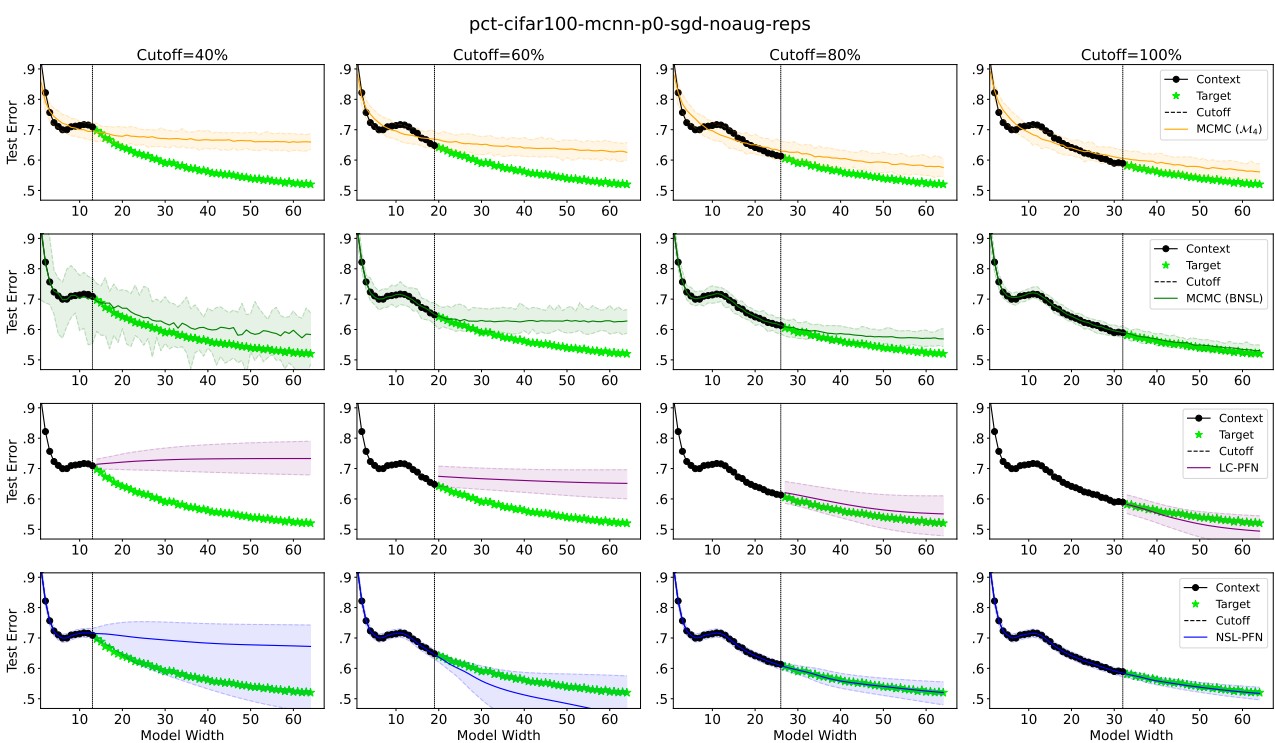

Figure 23: **Visualization of extrapolation results on double descent (DD; Nakkiran et al., 2021), Part 8 of 8.**

