# OpenReview forum: "Bayesian Neural Scaling Law Extrapolation with Prior-Data Fitted Networks"
_ICML.cc/2025/Conference — ICML 2025 poster_

### Official Review · Reviewer_uVCN · 2025-03-03

**Overall Recommendation:** 3

**Summary:**

The paper proposes using a taylored prior-fitted network to extrapolate neural scaling laws. A new prior over the set of potential curves is designed, capturing both simple power law behavior and more complex double-descent curves. The method is evaluated extensively on several data sets and shows improved performance over existing baselines.

## update after rebuttal

The rebuttal resolved some issues that I had and I raised my score accordingly.

**Claims And Evidence:**

The authors evaluate their new model extensively against relevant competitors and provide some ablation studies. However, I was left somewhat unconvinced. The main issue is that the researchers have so many degrees of freedom in building their model (and even in implementing some competitor methods) that it should be easy to craft a method that works better on an already available benchmark suite. Since the benchmark suite is relatively comprehensive and realistic, this may still be valuable.

**Essential References Not Discussed:**

None.

**Experimental Designs Or Analyses:**

I have some issues with how the MCM alternatives were implemented.
* The models $\Mcal_j$, BNSL appear to be specified without noise. That is, we expect the observed curves to follow *exactly* the idealized curves. This makes the Bayesian inference problem extremely misspecified, so it's no wonder it doesn't perform well. A more reasonable approach would be to do MCMC inference on the model $y = f(x) + \epsilon(x)$ where $\epsilon(x)$ is, e.g., a mean-zero Gaussian process.
* The number of samples drawn (150) is tiny. Typical numbers for running MCMC chains are in the thousands.

**Methods And Evaluation Criteria:**

Yes.

**Other Comments Or Suggestions:**

* Eq (10):
  * The expectation is over $\mathcal T$ which doesn't appear in the expression at all. Do you mean that $(Y^*, X^*)$ is one instance from $\mathcal T$?
  * I have no idea what $(Y, X)$ and "the autoregressive objective" are.

**Other Strengths And Weaknesses:**

The paper is really well written.

**Questions For Authors:**

To convince me about the scientific progress made in the papers, I strongly suggest the authors extend their empirical evaluation.
1. The MCMC baselines should be implemented in a more reasonable way, esp. regarding noise model and length of the chains; see my comments above.
2. The priors in Table 1 seem a bit arbitrary. How were they chosen? How sensitive are the results to their choice?

If these points are addressed convincingly I am inclined to increase my score. (Update: score raised to 3.)

**Relation To Broader Scientific Literature:**

The new model improves on several prior works for estimating the parameters in neural scaling laws. PFNs were already previously used for this task by [1]. The main innovation is to design priors tailored explicitly to the task and train a PFN accordingly.

[1] Adriaensen, S., Rakotoarison, H., Müller, S., & Hutter, F. (2023). Efficient bayesian learning curve extrapolation using prior-data fitted networks. Advances in Neural Information Processing Systems, 36, 19858-19886.

**Theoretical Claims:**

There are no theoretical claims.

---

> ### Author Rebuttal · Authors · 2025-03-31
>
> Thank you for your time and thoughtful comments. We address each of your comments below. If you have further questions, we’d be happy to discuss them during the author-reviewer interaction phase.
>
> ---
>
> > **[Q1]** The MCMC models appear to be specified without noise... This makes the Bayesian inference problem extremely misspecified.
>
> - Thank you for pointing this out, and sorry for the confusion. Actually, we **did account for the observational noise** in the MCMC baselines. Otherwise, we would not be able to compute the log-likelihood.
> - For example, $\mathcal{M}_1: y=ax^{-b}$ model has two parameters $a$ and $b$, and we define the likelihood as $\mathcal{N}(y; ax^{-b}, \sigma^2)$, where $\sigma^2$ is the observational noise.
> - MCMC is performed over the joint posterior: $p(a, b, \sigma \mid \lbrace x_i, y_i\rbrace_{i=1}^M) \propto  \prod_{i=1}^M \mathcal{N}(y_i; ax_i^{-b}, \sigma^2)  \cdot p(a) \cdot p(b) \cdot p(\sigma)$, where $p(a)$, $p(b)$, and $p(\sigma)$ are predefined priors.
> - We will clarify this in the revision to avoid further confusion. Thank you again for raising this point.
>
> ---
>
> > **[Q2]** The number of samples drawn (150) is tiny. Typical numbers for running MCMC chains are in the thousands.
>
> - Thank you for pointing this out. We acknowledge that 150 MCMC samples are small compared to the typical choice. We chose such a small number because there are too many scaling curves to evaluate and each MCMC inference takes too much time. For instance, applying 3000 MCMC samples to all those curves takes around **4 weeks** in total!
> - In [this external link](https://github.com/nslpfn-anonymous/nslpfn-anonymous/blob/rebuttal/rebuttal/LL_nsamples_and_inference_time.pdf), we compare our method against MCMC baselines with varying numbers of samples: 150, 300, 750, 1500, and 3000. To make the experiments feasible, we evaluated only on a subset of 30 scaling laws randomly sampled from the collection of all neural scaling law datasets we considered.
> - As shown in the results, increasing the number of MCMC samples sometimes leads to slightly improved performance as expected, but the differences are marginal (actually, we had already verified it before submission). Also, the increased performances still fall short of our method, while the inference time increases substantially. This supports one of our key claims: **NSL-PFN achieves strong performance with significantly better efficiency**. After meta-training, PFN-based methods provide **near-instantaneous inference**, which is the power of amortized inference.
> - We will discuss this further and provide the full evaluations in the revision.
>
> ---
>
> > **[Q3]** Eq (10): The expectation is over T which doesn't appear in the expression at all. Do you mean that (Y*, X*) is one instance from T? I have no idea what (X,Y) and "the autoregressive objective" are.
>
> - As defined in L123–124 (the right column) in the main paper, the context and target set are denoted as $\mathcal{C}=(X,Y)$ and $\mathcal{T} = (X^*, Y^*)$, respectively.
> - In Eq. (10), $\mathcal{C}, \mathcal{T}\sim p(\mathcal{D})$ is thus equivalent to $(X,Y),(X^*, Y^*)\sim p(\mathcal{D})$. We call the training objective $\mathbb{E}_{\mathcal{C}, \mathcal{T} \sim p(\mathcal{D})}  [\log q (Y \mid X, \mathcal{C})]$ "autoregressive" because it is regressing $\mathcal{C}=(X,Y)$ conditioned on the same context $\mathcal{C}$.
> - To improve clarity, we will revise the terminology to **"context regression objective"**, or any terminology you have in mind. Thank you for bringing this to our attention.
>
> ---
>
> > **[Q4]** The priors in Table 1 seem a bit arbitrary. How were they chosen? How sensitive are the results to their choice?
>
> - Thank you for pointing it out. For the design of our functional prior, we followed the same procedure used in LC-PFN. Specifically, we collected some of the well-known scaling law curves from various domains, and tried to match by eye the shape of our functional prior to the actual curves by manually controlling the parameters of the prior.
> - Importantly, when controlling the prior parameters, we assumed that the actual values of the scaling law curves are **not** accessible. Also, for the ColPret dataset (whose size exceeds the sum of all the other datasets), we did not access the shape of the scaling law curves, but just tested our model on the dataset directly. Nevertheless, the model still achieved strong performance on ColPret, indicating that our prior generalizes well across tasks.
> - Further, during the rebuttal period, we conducted a simple Bayesian optimization (BO) on the prior parameters to minimize the average RMSLE over 60 BO steps. As shown in [this external link](https://github.com/nslpfn-anonymous/nslpfn-anonymous/blob/rebuttal/rebuttal/hpo_error.pdf), the BO led to slight improvements on the performance, indicating that our prior tuning was not aggressive and that there remains room for further improvements.
> - We will discuss the above in the revision.

---

### Official Review · Reviewer_zhqD · 2025-03-12

**Overall Recommendation:** 4

**Summary:**

The authors propose a method to infer scaling laws using a Bayesian approach which allows them to model predictive uncertainty.
To accomplish this, they rely on prior-data fitted networks (PFNs) introduced by Müller et al., (2022) and their ability to meta-learn from large amounts of synthetic data.

The method is evaluated both quantitatively and qualitatively on a range of data sets showing competitive and improved performances against several baselines.

**Claims And Evidence:**

All claims (080 to 092) are provided with clear evidence throughout the paper.

**Essential References Not Discussed:**

All essential references are discussed properly.

**Experimental Designs Or Analyses:**

The experimental design is well-specified and well-motivated. The analysis is sound.

**Methods And Evaluation Criteria:**

The proposed method makes sense for the proposed task. PFNs and other meta-learning methods that learn from a large amount of synthetic data are an obvious fit for tasks such as neural scaling law inference, due to the ease of generating such synthetic data while at the same time having very limited real-world data.

The evaluation criteria are also broad and valid, both quantitatively, as well as qualitative visualizations.

**Other Comments Or Suggestions:**

- The reference section needs to be updated, as several references point to arxiv preprints despite the respective papers being published work. An example is Müller et al. (2021) which has been published at ICLR 2022.

**Other Strengths And Weaknesses:**

### Strengths
- The paper is very well written and can easily be followed.
- The motivation is clear and justified. The evaluation is convincing.


### Weaknesses
- One omission from the related work discussion in Appendix A is the model family of neural processes originally introduced by Garnelo et al. in 2018.
These follow a similar structure and motivation to PFNs and have been adopted in a wide area of applications.

**Questions For Authors:**

- Q1: The PFN has been trained with a rather large set of synthetic examples (1.6 million). How sensitive is the performance of the proposed approach to this size?

**Relation To Broader Scientific Literature:**

The relation to the broader scientific literature is in the application of PFNs for the task of scaling law inference.
The proposal does not contribute anything to the theory of PFNs but takes it as a tool to improve current neural scaling law inference techniques.

**Theoretical Claims:**

There are no theoretical claims that require proof.

---

> ### Author Rebuttal · Authors · 2025-03-31
>
> Thank you for your time and thoughtful comments. We address each of your comments below. If you have any further questions or comments, we would be happy to discuss them during the author-reviewer interaction phase.
>
> ---
> >**[Q1]** Missing related work - Neural Processes.
>
> - Thank you for the suggestion. We agree that the Neural Process (NP) family, originally introduced by Garnelo et al. [1], is closely related to our approach, as both aim to meta-learn an amortized posterior inference machine over a distribution of tasks. We will include the following discussion in the revision:
> - *Neural Processes (NPs) [1] typically learn a latent variable model where context data are summarized into a global latent representation, and target predictions are made by conditioning on this latent variable. This structure allows for uncertainty modeling and task generalization. The Conditional Neural Process (CNP) [2] simplifies this by removing the latent variable and directly conditioning target predictions on the aggregated context set. The Attentive Neural Process (ANP) [3] improves upon this by incorporating attention mechanisms to better capture context-target relationships. These models have been widely applied to few-shot learning, time series modeling, spatial inference, and so on. Further extensions include Transformer Neural Processes (TNPs) [4], which replace context aggregation with attention-based sequence modeling using Transformer architectures. While conceptually related, PFNs differ from NPs in an important way: PFNs do not require real dataset to train the inference machine. Instead, PFNs directly meta-learn the posterior predictive distribution with synthetic data generated from a manually-defined prior distribution, which allows Bayesian inference instead of maximum likelihood estimation. On the other hand, NPs are basically MLE and can be seen as implicitly learning the data-driven prior.*
>
> ---
>
> >**[Q2]** The reference section needs to be updated, as several references point to arxiv preprints despite the respective papers being published work.
>
> - Thank you for the suggestion. We will clarify the publication status of all references and replace arXiv citations with their corresponding published versions wherever applicable. We appreciate your attention to detail and will ensure the revision reflects the accurate and up-to-date citations as follows:
>
> - *Exploring the Limits of Large Scale Pre-Training* → ICLR 2022
> - *An Image Is Worth 16x16 Words: Transformers for Image Recognition at Scale* → ICLR 2021
> - *Language Models Are Few-Shot Learners* → NeurIPS 2020
> - *Broken Neural Scaling Laws* → ICLR 2023
> - *BERT: Pre-Training of Deep Bidirectional Transformers for Language Understanding* → NAACL 2019
> - *A Survey on In-Context Learning* → EMNLP 2024
> - *Probabilistic Rollouts for Learning Curve Extrapolation Across Hyperparameter Settings* → ICML Workshop 2019
> - *Scaling Laws for Neural Machine Translation* → ICLR 2022
> - *Mamba: Linear-Time Sequence Modeling with Selective State Spaces* → COLM 2024
> - *Training Compute-Optimal Large Language Models* → NeurIPS 2022
> - *TabPFN: A Transformer That Solves Small Tabular Classification Problems in a Second* → ICLR 2023
> - *Bayesian Active Learning for Classification and Preference Learning* → CoRR 2011
> - *Adam: A Method for Stochastic Optimization* → ICLR 2015
> - *Transformers Can Do Bayesian Inference* → ICLR 2022
> - *In-Context Freeze-Thaw Bayesian Optimization for Hyperparameter Optimization* → ICML 2024
> - ...
>
> ---
>
> >**[Q3]** The PFN has been trained with a rather large set of synthetic examples (1.6 million). How sensitive is the performance of the proposed approach to this size?
>
> - Thank you for the question. First of all, it is well-known that PFNs benefit from training on a large amount of synthetic data, as shown in LC-PFN and other related works. Since PFNs are trained to approximate the posterior predictive over a distribution of tasks, a large number of sampled functions is typically required for effective generalization.
> - In [this external link](https://github.com/nslpfn-anonymous/nslpfn-anonymous/blob/rebuttal/rebuttal/learning_curves.pdf), we provide the training losses and test performances over training iterations. We find that they keep converging as the model spends more synthetic data. Note that for each iteration, we randomly sample a batch of synthetic functions (which can be generated cheaply on-the-fly from our functional prior), hence the number of synthetic training data used is proportional to the training iterations.
>
> ---
>
> **Reference**
>
> [1] Garnelo et al. *Neural Processes. ICML workshop,* 2018.
>
> [2] Garnelo et al. *Conditional Neural Processes, ICML* 2018.
>
> [3] Kim et al. *Attentive Neural Processes, ICML* 2019.
>
> [4] Nguyen et al. *Transformer Neural Processes: Uncertainty-Aware Meta Learning via Sequence Modeling. ICML* 2022.

---

> > ### Comment · Reviewer_zhqD · 2025-04-02
> >
> > Thank your for your rebuttal and the answers which resolved my remaining questions/requests.

---

> > > ### Author Response · Authors · 2025-04-03
> > >
> > > We are glad to hear that your concerns have been resolved through this rebuttal. We appreciate your time and thoughtful feedback once again. Please let us know if you have any further questions, and we will be pleased to address them.
> > >
> > > Best regards,
> > >
> > > The Authors

---

### Official Review · Reviewer_ogXf · 2025-03-13

**Overall Recommendation:** 3

**Summary:**

This paper proposes the use of Bayesian Neural Networks (BNNs) to model deep learning scaling laws.
Scaling laws in deep learning have gained a considerable amount of interest recently, as they provide quantitative ways to estimate, for example, the amount of computational resources, model size, or dataset size needed to achieve certain performance.
Traditional approaches estimate these neural scaling laws using simple parametric assumptions (e.g., power law, exponential), and by carrying out "point estimation", that is without accounting for any uncertainty in predictions/extrapolations.
BNNs allow one to tackle these two limitations by employing flexible over-parameterized neural network models and by treating these in a Bayesian way.
The added benefit of this approach is the possibility to carry out active learning.
One of the main contributions is the use of Prior Fitted Networks (PFNs) to establish sensible neural scaling laws that can be generated from the model prior to conditioning on any data.

**Claims And Evidence:**

The claims are that the proposed approach gives flexible representation for a prior over neural scaling laws and that the proposed model and Bayesian treatment yield accurate extrapolations.
Another claim is that uncertainty in scaling laws can be used effectively to carry out active learning leading to computational savings.

These claims are generally supported by the experiments provided in the paper.
However, given that this paper is mostly experimental in nature, I would have liked to see a much more comprehensive validation, including more active learning scenarios and more competitors.
Also, I think it would have been useful to study the calibration of the predictive uncertainties; at the moment, it is not clear how well the uncertainties produced by the model are calibrated.

**Essential References Not Discussed:**

I don't think that there are other essential references to be added.

**Experimental Designs Or Analyses:**

As far as I can tell, the experimental design and analyses in the paper are conducted properly.

**Methods And Evaluation Criteria:**

The proposed methods and evaluation criteria are generally ok, but again I think it would be important to evaluate calibration, which is currently missing.

**Other Comments Or Suggestions:**

The paper is generally well written, so I don't have any particular comments on this.

**Other Strengths And Weaknesses:**

The main weakness is the formulation of a rather detailed PFN for a task that is essentially 1D regression/extrapolation.
My immediate reaction after reading this paper was that 1D regression can be done in closed form with Bayesian linear regression, and it can be made quite flexible through the use of basis functions.
This would give access to the marginal likelihood for model selection, and it would enable all the nice things that this paper promises to do quite cheaply.
I believe that this would be a good alternative to the approach proposed in this paper, and I would encourage the Authors to think about this type of competitor.
Section 3.1 provides a long text specifying the assumptions on the functions that BNNs can represent, whereas there is no effort in trying to encode this information in some basis functions which might ultimately yield competitive performance.

**Questions For Authors:**

Overall I think this paper has a good aim.
I find the realization overly complicated and I think that the paper is missing some simple Bayesian competitors.
Due to the paper being experimental in nature, I would encourage the Authors to focus their efforts on a wider experimental campaign, testing as many competitors as possible, and reporting calibration.

**Relation To Broader Scientific Literature:**

I think that the literature review on neural scaling laws is nicely captured in the paper, although I am not an expert on this.
Maybe I would have put a bit more emphasis on alternative approaches on BNNs beyond PFNs.

**Theoretical Claims:**

There are no theoretical developments in this paper.

---

> ### Author Rebuttal · Authors · 2025-03-31
>
> Thank you for your time and thoughtful comments. We address each of your comments below. If you have any further questions or comments, we would be happy to discuss them during the author-reviewer interaction phase.
>
>
> ---
>
> >**[Q1]** Additional experiments - calibration of the predictive uncertainties.
>
> - Thank you for the suggestion. In the following, we use **mean square calibration error (MSCE)**, the calibration metric used in [1] to assess predictive uncertainty. As shown in [this external link](https://github.com/nslpfn-anonymous/nslpfn-anonymous/blob/rebuttal/rebuttal/main_table_with_calibration_error.md), **our method demonstrates notably better calibration performances than the baselines**.
> - The gap is more prominent especially on the Double Descent (DD) which exhibits **highly chaotic behavior**. In such regimes, well-calibrated uncertainty is particularly important for reliable extrapolation. Furthermore, our method performs especially well on ColPret, the most recent and largest collection of **realistic neural scaling laws**, including models like GPT-3.
> - These results show our model’s strength in handling challenging and real-world scaling behaviors. We will include those findings in the revision.
>
> ---
>
> >**[Q2]** Additional baselines - Bayesian linear regression with basis functions.
>
> - Thank you for the suggestion. We agree that Bayesian linear regression (BLR) [2] with appropriate basis functions is a natural and lightweight approach for 1D regression and extrapolation.
> - To evaluate, we conducted additional experiments with the following baselines: 1) BLR with neural network basis functions, 2) BLR with polynomial basis, 3) BLR with RBF basis, 4) BLR with Fourier basis, 5) BLR with sigmoid basis, 6) BLR with spline basis, and 7) Deep Kernel GP (DKGP) [3]. We use log marginal likelihood for tuning the parameters, such as the parameters of the NN as a BLR basis function. We provide qualitative examples of the BLR with NN baseline and DKGP at [this external link](https://github.com/nslpfn-anonymous/nslpfn-anonymous/tree/rebuttal/rebuttal/BLR_visualization) and [this externel link](https://github.com/nslpfn-anonymous/nslpfn-anonymous/tree/rebuttal/rebuttal/DKGP_visualization), respectively.
> - As shown in [this external link](https://github.com/nslpfn-anonymous/nslpfn-anonymous/blob/rebuttal/rebuttal/table_with_dkgp_and_blr.md), **those baselines are not competitive** in the context of neural scaling laws. This is because, while they offer interpretability and closed-form inference, to our knowledge **it is not straightforward to encode a set of inductive biases about neural scaling laws into the basis or kernel functions**. This is why we originally did not consider those baselines.
> - In contrast, our PFN-based method provides a flexible framework to encode our prior belief on the shape of the functions. We can use any functional prior as long as we can efficiently sample from it, which is one of the biggest advantages of using PFNs. This is precisely why we propose to use PFNs for the problem of extrapolating neural scaling laws.
> - We appreciate your further suggestions as to how to effectively encode the prior belief on the scaling law behaviors into the corresponding basis or kernel functions. As long as time allows, we will try our best to do the additional experiments following your suggestions.
>
> ---
>
> >**[Q3]** I would have liked to see a much more comprehensive validation.
>
> - We emphasize that, to the best of our knowledge, we have already collected **all publicly available datasets of neural scaling laws**, including recent and diverse settings such as **ColPret** (large-scale LLMs), **DD** (chaotic double descent behavior), and other benchmarks from $\mathcal{M}_4$ and BNSL. These datasets span a wide range of domains and scaling patterns, including smooth, non-smooth, and chaotic regimes.
> - Across all of these settings, our method consistently shows **strong performance**, demonstrating its general applicability.
> - Also, during the rebuttal period, we conducted the two additional experiments following your suggestion. 1) We additionally consider Bayesian linear regression and deep kernel GP baselines and find that they largely underperform our method. 2) We demonstrate that our method shows superior calibration performance than the baselines.
> - Our paper also includes Bayesian active learning experiments to demonstrate the usefulness of uncertainty, which has not been discussed in the LC-PFN paper. If you find it insufficient, please feel free to suggest any additional experiments in detail, and we would be happy to try our best to do the experiments as long as time allows.
>
> ---
>
> **References**
>
> [1] Kuleshov et al., *Accurate Uncertainties for Deep Learning Using Calibrated Regression. ICML* 2018.
>
> [2] Bishop et al., *Pattern Recognition and Machine Learning*, 2006.
>
> [3] Wilson et al., *Deep Kernel Learning, AISTATS* 2016

---

### Official Review · Reviewer_1HYA · 2025-03-13

**Overall Recommendation:** 3

**Summary:**

In this paper, the authors investigate the use of Prior-Data Fitted Networks (PFNs), a Bayesian framework for estimating neural scaling law curves. Existing methods typically produce point estimates, which fail to capture the true scaling law dynamics across different scenarios (e.g., double descent, flat cut-offs) and do not provide uncertainty estimates. These uncertainties are crucial for designing experiments based on early observations of the scaling curve. To address this, the authors introduce a family of priors that effectively represent scaling laws in large model training, synthesizing new samples from these priors to train PFNs, which are based on a standard transformer architecture. The authors demonstrate that their approach, referred to as NSL-PFNs, not only quantifies Bayesian uncertainty but also yields strong performance in terms of RMSLE and likelihood metrics. Extensive evaluations across image and text domain benchmarks, along with ablation studies, highlight the contributions of each component in their framework.

**Claims And Evidence:**

Yes, the claims are supported by relevant evidence.

**Essential References Not Discussed:**

N/A

**Experimental Designs Or Analyses:**

Yes, the experimental setup and design is sound.

**Methods And Evaluation Criteria:**

Yes, the authors have considered relevant baselines and benchmarks datasets.

**Other Comments Or Suggestions:**

N/A

**Other Strengths And Weaknesses:**

In my view, the primary contribution of this paper lies in the development of a novel family of priors that enhance the training of PFNs, which constitutes the core strength of the work. Other aspects of the paper, as cited by the authors, have been explored in prior research. The emphasis on Bayesian uncertainty in the context of scaling law estimation, given only a set of observations, is particularly valuable for practitioners. It can eventually provide credible intervals and practical insights for conducting experiments at varying scales. Additionally, the paper is exceptionally well-written, clear, and almost free of typos.

**Questions For Authors:**

Can the authors clarify the main novelty of this work compared to Adriaensen et al.? Is it primarily the definition of the functional priors? It would be beneficial to explicitly highlight both high- and low-level differences.
In Table 1, can the authors provide a reference that demonstrates the priors used for the coefficients (e.g., $\log a \sim \mathcal{N}(-1, 0.6)$)? How do these choices impact the overall performance?
While the authors follow Muller et al. for implementing the PFN, can they clarify why positional encoding has been omitted?
Can the authors update Figure 4 to better highlight the uncertainties for each method in the extrapolation regime? Currently, it is difficult to interpret and extract insights from the figure.

**Relation To Broader Scientific Literature:**

In my humble opinion, the contributions are significant within the broader context of the scientific literature. However, there are some questions I have raised that require further clarification.

**Theoretical Claims:**

N/A

---

> ### Author Rebuttal · Authors · 2025-03-31
>
> Thank you for your time and thoughtful comments. We address each of your comments below. If you have any further questions or comments, we would be happy to discuss them during the author-reviewer interaction phase.
>
>
> ---
>
> >**[Q1]** Main novelty of this work compared to LC-PFN at both high- and low-level?
>
> Thank you for the question. The main novelty of our work compared to LC-PFN [1] lies in both the high-level approach and low-level modeling choices:
>
> High-level differences:
> - Our work is the first to apply a **Bayesian** framework to the problem of **neural scaling law extrapolation**, whose functional distribution and characteristics are much different from those of learning curves the LC-PFN focused on.
> - We further leverage uncertainty estimates to actively guide data acquisition, demonstrating the **practical utility of uncertainty in reducing the cost of collecting high-quality scaling data** — something not explored in LC-PFN.
>
> Low-level differences:
> - As you mentioned, we carefully design **a novel functional prior tailored to neural scaling laws**, informed by limitations of previous empirical approaches. Our prior can capture complex and chaotic behaviors such as double descent, enabling accurate extrapolation and reliable uncertainty estimates even under challenging real-world scenarios. In contrast, the prior of LC-PFN does not cover such complex patterns.
> - We further propose a simple **context-regressive learning objective** that not only improves extrapolation performance but also enables Bayesian active learning — something that LC-PFN lacks.
>
> We will make these distinctions more explicit in the revision.
>
> ---
>
> >**[Q2]** In Table 1, can the authors provide a reference that demonstrates the priors used for the coefficients? How do these choices impact the overall performance?
>
> - Thank you for pointing it out. For the design of our functional prior, we followed the same procedure used in LC-PFN [1]. Specifically, we collected some of the well-known scaling law curves from various domains, and tried to match by the eye the shape of our functional prior to the actual curves by manually controlling the parameters of the prior.
> - Importantly, when controlling the prior parameters, we have assumed that the actual values of the scaling law curves are **not** accessible. Also, for the ColPret dataset (whose size exceeds the sum of all the other datasets), we did not even access the shape of the scaling law curves, but just test our model on the dataset directly. Nevertheless, the model still achieved strong performance on ColPret, indicating that our prior generalizes well across tasks.
> - Further, during the rebuttal period, we conducted a simple Bayesian optimization (BO) on the prior parameters to minimize the average RMSLE over 60 BO steps. As shown in [this external link](https://github.com/nslpfn-anonymous/nslpfn-anonymous/blob/rebuttal/rebuttal/hpo_error.pdf), the BO led to slight improvements on the performance, which indicates that our procedure of tuning prior parameters was not aggressive and there remain some rooms for further improvements of our prior.
> - We will discuss the above in the revision.
>
> ---
>
> >**[Q3]** While the authors follow Muller et al. for implementing the PFN, can they clarify why positional encoding has been omitted?
>
> - The original PFN paper (Muller et al. [2]) also omits the positional encoding for the following two reasons.
> - In PFNs, when encoding the context points $\mathcal{C}=\lbrace(x_i,y_i)\rbrace_{i=1}^M$, the input feature $x_i$ itself already contains the "positional information" in the input space.
> - Further, PFNs approximate the posterior predictive distribution $p(y^*,x^*|\mathcal{C})$, which is permutation-invariant w.r.t. the ordering of the context points $\mathcal{C}=\lbrace(x_i,y_i)\rbrace_{i=1}^M$, by definition. The same applies to our model as well as LC-PFN [1].
> - We will clarify this in the revision.
>
> ---
>
> >**[Q4]** Update Figure 4 to better highlight the uncertainties for each method?
>
> - Thank you for the suggestion. To improve clarity, we have updated those figures by separating the uncertainty plots for each method. [This external link](https://github.com/nslpfn-anonymous/nslpfn-anonymous/tree/rebuttal/rebuttal/DD_visualization) includes all those visualizations. We will include them in the appendix of the revision.
>
> ---
>
> **References**
>
> [1] Adriaensen et al., *Efficient Learning Curve Extrapolation via Prior-Data Fitted Networks, NeurIPS* 2023.
>
> [2] Müller et al., *Transformers Can Do Bayesian Inference, ICLR* 2022

---

### Decision · Program_Chairs · 2025-05-01

**Decision:**

Accept (poster)

**Comment:**

This paper concerns fitting the learning curves from scaling-up neural network experiments as a way to explain the neural scaling law. The authors proposed a Bayesian approach, where contributions include (1) design of the candidate curve families and principles for function prior construction, (2) prior-fitted networks which fits on data generated from the function prior in (1) using a (variational Bayesian) meta-learning style training objective, and (3) the trained PFN will then allow uncertainty estimation for the extrapolation region when tested on its fitness regarding scaling curves from real-word neural network scaling experiments. Experiments on benchmark scaling law curve fitting datasets show the method performs better regarding extrapolation when compared with deterministic methods and simple MCMC baselines over the parameters of pre-specified curve families.

Reviewers found the work to be a novel application of Bayesian methods + meta-learning approaches to the analysis of neural scaling law curves. However, reviewers also raised questions regarding the choice of the fitting model (prior-fitted network vs Bayesian linear regression with non-linear 1D features) as well as the uncertainty quality. Author rebuttal has address some but not most of the questions.

I briefly read the paper.
- From the point of view of designing the algorithm for fitting the PFN as well as uncertainty quantification, I personally think these approaches are fairly straightforward to people coming from Bayesian neural networks and meta-learning/neural process fields.
- However I think another strength of the paper is the careful design of the functional prior, which demonstrates a "Bayesian" spirit very well, in the sense that it recognises existing approaches and take inspirations from them as prior knowledge, and the authors also made careful design choices regarding different parts of the scaling law curve.
- Still I would like to see an MCMC baseline where the functional prior is the designed prior in section 3.1, rather than the parametric prior over each of the function class (say M1 or M2). This can ablate the need of a prior-fitted network. The authors might argue this prior is difficult to perform exact posterior inference (via MCMC). However, I feel this is a hierarchical prior but still parametric, so perhaps methods like a combination of Gibbs and Langevin samplers can be useful.

Overall, I think this paper would be more meaningful to present to neural scaling law community, in a way to show an alternative approach to "guess" how neural networks scale :)